# Flow Matching for Tabular Data Synthesis

**Bahrul Ilmi Nasution**[*]                                                              *bahrul.nasution@manchester.ac.uk*
*Department of Social Statistics*
*The University of Manchester*

**Floor Eijkelboom**                                                                         *f.eijkelboom@uva.nl*
*Amsterdam Machine Learning Lab*
*University of Amsterdam*

**Mark Elliot**                                                                          *mark.elliot@manchester.ac.uk*
*Department of Social Statistics*
*The University of Manchester*

**Richard Allmendinger**                                                      *richard.allmendinger@manchester.ac.uk*
*Alliance Manchester Business School*
*The University of Manchester*

**Christian A. Naesseth**                                                                  *c.a.naesseth@uva.nl*
*Amsterdam Machine Learning Lab*
*University of Amsterdam*

**Reviewed on OpenReview:** *https://openreview.net/forum?id=RdOjoAa66L*

## Abstract

Synthetic data generation is an important tool for privacy-preserving data sharing. Although diffusion models have set recent benchmarks, flow matching (FM) offers a promising alternative. This paper presents different ways to implement FM for tabular data synthesis. We provide a comprehensive empirical study that compares flow matching (FM and variational FM) with a state-of-the-art diffusion method (TabDDPM and TabSyn) in tabular data synthesis. We evaluate both the standard Optimal Transport (OT) and the Variance Preserving (VP) probability paths, and also compare deterministic and stochastic samplers – something possible when learning to generate using *variational* FM – characterising the empirical relationship between data utility and privacy risk. Our key findings reveal that FM, particularly TabbyFlow, outperforms diffusion baselines. Flow matching methods also achieve better performance with remarkably low function evaluations ($\leq 100$ steps), offering a substantial computational advantage. The choice of probability path is also crucial, as using the OT is a strong default and more robust to early stopping on average, while VP has potential to produce synthetic data with lower privacy risk. Lastly, our results show that making flows stochastic not only preserves marginal distributions but, in some instances, enables the generation of high utility synthetic data with reduced disclosure risk. The implementation code associated with this paper is publicly available at `https://github.com/rulnasution/tabular-flow-matching`.

## 1 Introduction

Government bodies, particularly national statistical offices (NSOs), are faced with a pressing challenge: how to disseminate useful tabular data while preserving individual privacy. Tabular data, which are structured

---

[*]Corresponding author. Part of this work was carried out during a research visit to University of Amsterdam.

data with rows and columns representing real-world entities and their attributes, are essential for decision-making in economics, healthcare, social sciences, and finance (Little et al. 2024). However, because these datasets often contain sensitive attributes, such as financial conditions or medical diagnoses, privacy protection laws such as the EU's General Data Protection Regulation (GDPR) and Indonesia's Personal Data Protection (PDP) Act limit access to them (Ruggles & Van Riper 2021). This tension between data utility and confidentiality has spurred interest in methods that can generate synthetic tabular data that preserve statistical properties while reducing the risk of disclosure (Nowok et al. 2016).

Recent advances in deep generative models have provided a promising approach to tabular data synthesis. Deep learning techniques, such as Generative Adversarial Networks (Goodfellow et al. 2014), Variational Autoencoders (Kingma & Welling 2013), and more recently, diffusion models (Sohl-Dickstein et al. 2015; Ho et al. 2020), have demonstrated impressive capabilities in generating synthetic data of various forms, including images and text, as well as tabular data (Xu et al. 2019; Kotelnikov et al. 2023).

Diffusion models, in particular, have emerged as a powerful option for generative tasks, including the synthesis of tabular data. TabDDPM (Kotelnikov et al. 2023), a diffusion model adapted for tabular data, and other models such as TabSyn (Zhang et al. 2024) have gained attention for their ability to model intricate dependencies between variables in tabular datasets by iteratively transforming noise into structured data. The main strength of these models lies in their ability to handle the high-dimensional structure of tabular data, where the relationships between variables are often nonlinear and complex.

Another recent development is flow matching (FM) (Lipman et al. 2023; Albergo et al. 2023; Liu 2022), which approximates the data distribution through a flow, i.e. through learning a velocity field that induces an ordinary differential equation transporting noise to data. Variational Flow Matching (VFM) (Eijkelboom et al. 2024) offers a reinterpretation of flow matching as a form of variational inference over trajectories, providing an elegant solution to parameterise the flow towards any kind of distribution (e.g. also discrete or constrained). This method holds significant promise for tabular data synthesis, as it ensures that the generated output not only mirrors the statistical attributes of the original data but also captures patterns that traditional generative techniques might overlook. Beyond tabular data generation (Guzmán-Cordero et al. 2025), VFM has also seen recent success in fields such as molecular generation (Zaghen et al. 2025b; Eijkelboom et al. 2025; Sakalyan et al. 2025), image generation (Matişan et al. 2025), and climate modeling (Finn et al. 2025). Further discussion of related work can be found in Appendix A.

Although recent advances in tabular diffusion models (Kotelnikov et al. 2023; Zhang et al. 2024) and FM (Guzmán-Cordero et al. 2025) have shown promise, existing studies remain limited in four ways. First, they have focused primarily on open source benchmark datasets, neglecting the unique challenges of census data as a real-world representation. Second, utility and privacy risk evaluation has been restricted to computer science metrics such as ML accuracy and distance to closest record (DCR), which (i) may not align with statistical needs,(ii) are computationally intensive, and (iii) can be hard to interpret for practitioners. Third, recent flow matching studies (Eijkelboom et al. 2024; Guzmán-Cordero et al. 2025; Eijkelboom et al. 2025) have predominantly focused on optimal transport trajectories. The potential of variance-preserving trajectories—although theoretically established in the original FM formulation (Lipman et al. 2023) and proven effective in diffusion models (Ho et al. 2020)—remains underexplored. Finally, the interaction between latent vs. direct representations and deterministic vs. stochastic dynamics in FM-based tabular synthesis remains unclear, leaving open questions about which configuration produces the best synthetic data.

**Research Objective.** The primary goal of this paper is to quantify whether flow matching can offer an efficient, high-utility, and low-risk alternative to diffusion models for mixed-type tabular data. We address the goal through four specific research questions.

- **Q1.** Can FM-based models improve upon diffusion model baselines in tabular data synthesis? (Section 5.1)

- **Q2.** How efficient are FM models in terms of the utility and risk of the generated output? (Section 5.2)

- **Q3.** How do utility and risk characteristics of the synthetic data evolve with integration time? (Section 5.3)

- **Q4.** Can stochastic sampling improve synthesis quality relative to deterministic sampling? (Section 5.4)

**Contributions.** We explore flow matching techniques for tabular data synthesis through four implementations: (1) learning the distribution of latent variables using regular flow matching, inspired by TabSyn (Zhang et al. 2024); (2) learning the tabular data distribution directly using variational flow matching; (3) a systematic comparison of interpolation schemes; and (4) implementing stochastic dynamics in VFM.

At the application level, our work bridges flow matching with the operational needs of statistical agencies by providing a systematic analysis of the utility and risk of the synthetic data under different experimental conditions, recognising that algorithms optimising fidelity may implicitly amplify disclosure risk. The results demonstrate that flow matching can generate high-quality synthetic tabular data while protecting against statistical disclosure, thus offering a new option for responsible data generation.

## 2  Background

### 2.1  Flow Matching

Flow matching (FM) (Lipman et al. 2023; Liu et al. 2022; Albergo et al. 2023) is a simulation-free generative modelling framework that learns a velocity field $v_t^\theta$ parameterised by $\theta \in \mathbb{R}^p$. This field defines a continuous transformation of a base distribution $p_0$ (typically a standard Gaussian) into a target distribution $p_1$ (empirical data) over a pseudo-time interval $t \in [0, 1]$ through its induced ordinary differential equation.

Rather than directly estimating the target velocity field $u_t(x)$ – which we do not have access to – FM defines conditional distributions $p_t(x_t \mid x_1)$ that make an assumption on the dynamics *towards a fixed* endpoint $x_1$, for which the *conditional* velocity is simple to compute. Under this formulation, the marginal velocity field is given by:

$$u_t(x_t) = \int u_t(x_t \mid x_1) \frac{p_t(x_t \mid x_1) p_{\text{data}}(x_1)}{p_t(x_t)} \, dx_1, \tag{2.1}$$

where $u_t(x_t \mid x_1)$ is the conditional velocity field. To avoid the computational cost of evaluating the integral in equation 2.1, FM learns the target velocity field through a Monte Carlo estimate of the conditional velocity, making the problem tractable. This approach is known as Conditional Flow Matching (CFM):

$$\mathcal{L}_{\text{CFM}}(\theta) = \mathbb{E}_{t \sim [0,1], x_1 \sim p_{\text{data}}, x_t \sim p_t(x_t \mid x_1)} \left[ \left\| v_t^\theta(x_t) - u_t(x_t \mid x_1) \right\|_2^2 \right]. \tag{2.2}$$

A common way to define this conditional distribution is through interpolation, i.e. define $x_t$ as a linear combination of $x_1$ and a noise sample $x_0$:

$$x_t = \alpha_t x_1 + \sigma_t x_0 \sim p_t(x_t \mid x_1) \quad \text{for} \quad x_0 \sim p_0, \ x_1 \sim p_1, \ t \in [0, 1]. \tag{2.3}$$

where $\alpha_t$, $\sigma_t$ are time-dependent coefficients that specify the trajectory. We can hence write down the corresponding conditional velocity:

$$u(x_t \mid x_1) = \dot{\alpha}_t x_1 + \dot{\sigma}_t \frac{x_t - \alpha_t x_1}{\sigma_t} = \underbrace{\left( \dot{\alpha}_t - \frac{\dot{\sigma}_t}{\sigma_t} \alpha_t \right) x_1}_{A_t(x_t)} + \underbrace{\frac{\dot{\sigma}_t}{\sigma_t} x_t}_{B_t(x_t)}. \tag{2.4}$$

Here, $A_t(x_t)$ and $B_t(x_t)$ are time-dependent coefficients that control the contributions of $x_1$ and $x_t$, respectively. This expression allows for efficient computation of the conditional velocity field without the need to perform numerical integration or simulation.

### 2.2  Variational Flow Matching

Although flow matching is already an efficient and simulation free generative modelling framework, it can struggle with multimodal data or data of heterogeneous types (Zhai & Hao 2025). To address this limitation,

Eijkelboom et al. (2024) proposed variational flow matching (VFM), which reinterprets flow matching using the perspective of variational inference.

Unlike standard FM, which is designed primarily for continuous data, VFM allows for a more flexible learning mechanism that accommodates a wider range of data types, including tabular data (Guzmán-Cordero et al. 2025). VFM learns an approximate vector field via the posterior distribution of the data using a variational distribution $q_t^\theta$:

$$v_t^\theta(x_t) := \int u_t(x_t \mid x_1) q_t^\theta(x_1 \mid x_t) \, dx_1. \tag{2.5}$$

This formulation allows the model to flexibly approximate the true conditional velocity while allowing the incorporation of data-specific constraints (e.g. discrete data) through the choice of posterior. The training objective in VFM is to minimise the divergence between the true joint distribution and the variational joint distribution, measured in the Kullback-Leibler divergence (KLD):

$$\mathcal{L}_{\text{MFVFM}}(\theta) = \mathbb{E}_t \left[ \text{KL} \left( p_t(x_t) p_t(x_1 \mid x_t) \,\big\|\, p_t(x_t) q_t^\theta(x_1 \mid x_t) \right) \right] = -\mathbb{E}_{t,x_1,x_t} \left[ \sum_{d=1}^D \log q_t^\theta(x_1^d \mid x_t) \right] + \text{const} \tag{2.6}$$

VFM uses mean-field variational inference, i.e., $q_t^\theta(x_1 \mid x_t) = \prod_{d=1}^D q_t^\theta(x_1^d \mid x_t)$, which corresponds to assuming conditional independence across dimensions under the variational posterior $q_t^\theta$ (not under the true posterior $p_t(x_1 \mid x_t)$). Following Eijkelboom et al. (2024), this choice produces a tractable objective and still enables recovery of the true distribution at $t = 1$. The mean field formulation also supports flexible variational distributions, particularly exponential families (Guzmán-Cordero et al. 2025) such as Gaussian or categorical distribution. Having learned the posterior distribution approximation, the vector field can be computed using equation 2.7:

$$v_t^\theta(x_t) = \mathbb{E}_{q_t^\theta(x_1 \mid x_t)} [u_t(x_t \mid x_1)] = \left( \dot{\alpha}_t - \frac{\dot{\sigma}_t}{\sigma_t} \alpha_t \right) \theta_t(x_t) + \frac{\dot{\sigma}_t}{\sigma_t} x_t \tag{2.7}$$

where $\theta_t(x_t)$ is the mean of the variational distribution $q_t^\theta$. Crucially, this formulation recovers standard flow matching in the Gaussian case.

**Stochastic Dynamics.** Beyond the deterministic ODE formulation, VFM also admits a stochastic counterpart that preserves the same marginal distribution $p_t(x)$ that can be approximated using variational distribution (Eijkelboom et al. 2024):

$$dx_t = \left[ v_t^\theta(x_t) + \frac{g_t^2}{2} s_t^\theta(x_t) \right] dt + g_t \, dw_t, \tag{2.8}$$

where $g_t \geq 0$ is a diffusion scheduler and $w_t$ is a standard Wiener process. The scheduler controls the stochasticity of the trajectory, but not the marginal path; Setting $g_t \equiv 0$ recovers the ODE. The drift is calculated using equation 2.7 and score $s_t^\theta(x_t)$ is estimated as:

$$s_t^\theta(x_t) = \mathbb{E}_{q_t^\theta(x_1 \mid x_t)} [\nabla_{x_t} \log p_t(x_t \mid x_1)] = -\frac{x_t - \alpha_t \theta_t(x_t)}{\sigma_t^2}. \tag{2.9}$$

This formulation supports approximate stochastic flow models, enabling richer models while still leveraging the variational structure for tractable learning.

## 3 Designing Flow Matching for Tabular Data Synthesis

### 3.1 Motivation and Design Axes

Flow Matching provides a general recipe for learning deterministic or stochastic transformations between probability distributions. However, its effectiveness in practice depends not on a single equation, but on

a set of design decisions that determine how distributions are represented, interpolated, and evolved over time. When applied to *tabular data synthesis*, these decisions interact in particularly complex ways. First, mixed continuous and categorical features require explicit representation choices, and moving to a continuous latent space can simplify the flow dynamics but does not remove the categorical challenge because discrete structure must still be captured and decoded. Second, privacy-sensitive settings impose computational limits, and statistical agencies often require explainable and reproducible model configurations. This section therefore reformulates Flow Matching as a *design space* rather than a single algorithmic recipe, and places tabular synthesis at its centre.

Our aim is to identify how four core axes – **representation**, **learning target**, **trajectory**, and **dynamics** – jointly shape the behaviour of tabular flow models.

1. **Representation** decides whether learning occurs in a continuous *latent space* that simplifies mixed data, or directly in the *data space*, preserving semantics.

2. **Learning target** distinguishes *Conditional Flow Matching* (velocity regression) from *Variational Flow Matching* (posterior regression), reflecting different ways of estimating the transport field.

3. **Trajectory** defines the interpolant between source and data distributions – commonly *Optimal Transport (OT)* or *Variance Preserving (VP)* paths – which control how signal and noise evolve over time.

4. **Dynamics** determine whether the learned flow is integrated deterministically as an *ODE* or stochastically as an *SDE*, corresponding to a deterministic transport map versus a stochastic process.

By systematically combining these axes, we focus on two complementary formulations. First, we introduce **TabSynFlow**, which learns deterministic ODE flows in latent space, substituting the diffusion process in TabSyn with a learned velocity field. We also evaluate **TabbyFlow** developed by Guzmán-Cordero et al. (2025) which extends Variational Flow Matching directly to data space, supporting both numerical and categorical variables through hybrid Gaussian-categorical posteriors.

We consider a dataset of mixed-type tabular observations $x = (x_{\text{num}}, x_{\text{cat}})$, where $x_{\text{num}} \in \mathbb{R}^{D_{\text{num}}}$ are continuous features and $x_{\text{cat}} \in \{1, \ldots, K_d\}^{D_{\text{cat}}}$ are categorical variables. The empirical data distribution is denoted by $p_1(x)$, while a simple prior $p_0(x)$, typically an isotropic Gaussian, serves as the source distribution. The goal of flow matching is to learn a continuous transformation that maps $p_0$ to $p_1$ through a parameterised time-dependent velocity field $v_t^\theta(x_t)$ defined on the interpolation in equation 2.3. The coefficients $(\alpha_t, \sigma_t)$ specify the probability path (for example, OT or VP trajectories) and induce a family of intermediate marginals $p_t(x_t)$. The generative process is recovered by integrating a deterministic ordinary differential equation (ODE) or a stochastic differential equation (SDE) via equation 2.8. The two instantiations below, TabbyFlow and TabSynFlow, correspond to different modelling choices for the representation space and the learning target.

## 3.2 TabSynFlow: Flow Matching in Latent Spaces

TabSynFlow extends latent-space generative modelling by replacing the diffusion component in TabSyn with a deterministic probability flow trained via Conditional Flow Matching (CFM). The model learns a time-dependent velocity field that transports a simple prior to the target latent distribution through an ODE, enabling a direct and continuous transformation during both training and sampling. This deterministic integration stabilises training and accelerates synthesis relative to iterative stochastic denoising, while retaining high-fidelity latent reconstructions through the encoder–decoder pathway.

TabSynFlow follows a two-stage framework. In the first stage, a VAE maps the raw tabular data, comprising both continuous and categorical features, into a continuous latent space. Subsequently, flow matching is applied to model the latent distribution $p(z_1)$, replacing the score-based diffusion model used in TabSyn. Gaussian noise samples $z_0 \sim \mathcal{N}(0, I)$ are continuously transformed into latent variables $z_1$ through an ODE parameterised by $v_t^\theta(z_t)$. The conditional velocity $u_t(z_t \mid z_1)$ is calculated using equation 2.4, and the loss becomes

$$\mathcal{L}_{\text{TabSynFlow}}(\theta) = \mathbb{E}_{t, z_1, z_t} \left[ \|v_t^\theta(z_t) - u_t(z_t \mid z_1)\|_2^2 \right]. \tag{3.1}$$

After training, the sampling proceeds by drawing $\tilde{z}_0 \sim \mathcal{N}(0, I)$ and integrating the learned ODE until $t = 1$. The resulting latent representation $\tilde{z}_1$ is then decoded to obtain a synthetic record $\tilde{x}$, followed by inverse transformation to the original feature space. Algorithms 1–2 in Appendix C.1 describe the procedure in detail.

Operating in latent space gives TabSynFlow efficient and stable training while retaining the expressive power of flow-based models. It supports fast, deterministic sampling and interpretable trajectories, making it a computationally efficient alternative to diffusion-based approaches for tabular data synthesis.

### 3.3 TabbyFlow: Variational Flow Matching in Data Space

TabbyFlow (Guzmán-Cordero et al. 2025) extends the Variational Flow Matching (VFM) framework of Eijkelboom et al. (2024) to structured tabular data containing numerical and categorical features. This formulation enables direct modelling in the data space, avoiding the need for an intermediate latent representation, while explicitly accounting for heterogeneous variable types within the flow dynamics.

The TabbyFlow objective treats each feature type according to its statistical nature: continuous variables are modelled using a Gaussian distribution, while categorical variables are modelled using categorical distributions. Let $x = (x_{\text{num}}, x_{\text{cat}})$ denote a sample comprising $D_{\text{num}}$ numerical and $D_{\text{cat}}$ categorical features. Each categorical feature is one-hot encoded, yielding total dimensionality $D = D_{\text{num}} + \sum_{d=1}^{D_{\text{cat}}} K_d$, where $K_d$ is the number of categories for the $d$-th categorical variable.

**Numerical Variables.** Following Theorem 3 of Eijkelboom et al. (2024), we assume a mean-field variational posterior $q_t^\theta(x_{\text{num},1} \mid x_t)$ with Gaussian structure parameterised by mean $\theta_t(x_t)$ and variance $0.5A_t(x_t)^{-2}$. The VFM loss for numerical variables is

$$\mathcal{L}_{\text{VFM-num}}(\theta) = \mathbb{E}_{t,x_t,x_1} \left[ \|A_t(x_t)(x_{\text{num},1} - \theta_t(x_t))\|_2^2 \right]. \tag{3.2}$$

However, we found that using $0.5A_t(x_t)^{-2}$ as variance yielded a lower performance in our experiment. To mitigate this, we implemented a relaxed variance of $0.5A_t(x_t)^{-1}$. This adjustment slightly inflates the variance, therefore acting as an empirical stabilisation tweak. The empirical results can be seen in Appendix G.1.

**Categorical Variables.** For categorical features, TabbyFlow models the variational posterior as a product of categorical distributions. Let $\theta_t^{dk}(x_t)$ denote the predicted probability of category $k$ in the $d$-th feature, obtained via softmax in the output layer. The loss minimises the cross-entropy with the one-hot targets $x_{\text{cat},1}^d$:

$$\mathcal{L}_{\text{VFM-cat}}(\theta) = -\mathbb{E}_{t,x_t,x_1} \left[ \sum_{d=1}^{D_{\text{cat}}} \sum_{k=1}^{K_d} \mathbb{I}[x_{\text{cat},1}^d = k] \log \theta_t^{dk}(x_t) \right]. \tag{3.3}$$

**Unified Objective.** Combining both loss terms yields the overall TabbyFlow objective:

$$\begin{aligned} \mathcal{L}_{\text{TabbyFlow}}(\theta) &= L_{\text{VFM-num}}(\theta) + L_{\text{VFM-cat}}(\theta) \\ &= \mathbb{E}_{t,x_t,x_1} \left[ \|\sqrt{A_t(x_t)}(x_{\text{num},1} - \theta_t(x_t))\|_2^2 \right] - \mathbb{E}_{t,x_t,x_1} \left[ \sum_{d=1}^{D_{\text{cat}}} \sum_{k=1}^{K_d} \mathbb{I}[x_{\text{cat},1}^d = k] \log \theta_t^{dk}(x_t) \right] \end{aligned} \tag{3.4}$$

**Sampling and Dynamics.** During sampling, a noise sample $\tilde{x}_0$ is drawn from $p_0$, and the trained network predicts $\theta_t(x_t)$ at each time step. Because TabbyFlow does not directly output a velocity field, it is recovered via equation 2.7, while the score function $s_t^\theta(x_t)$ via equation 2.9. These components define the ODE or SDE dynamics used for synthesis (Algorithms 3–4 in Appendix C.2). TabbyFlow thus enables a unified treatment of numerical and categorical features within a single variational flow, generalising continuous-time generative modelling to structured tabular domains.

# 4 Experimental Setting

## 4.1 Data

Table 1 summarises the datasets used in our empirical evaluation. Following Ran et al. (2024), Little et al. (2024), and Nasution et al. (2026), we used four census datasets (from the UK, Fiji, Canada and Rwanda), plus one additional census data from Indonesia. We also added two additional databases from UCI and Kaggle, which are standard benchmarks for tabular data (Kotelnikov et al. 2023). Unlike Ran et al. (2024), we treat age as continuous to exploit its natural ordering. Also, this choice remains compatible with TabDDPM/TabSyn, which operate on mixed-type data. To ensure a consistent and fair comparison, we adopt the same convention for TabSynFlow and TabbyFlow. Although these are our core benchmark datasets, we also provide a stress test of TabSynFlow and TabbyFlow on high cardinality and dimensional data in Appendix G.2.

## 4.2 Model and Environments

To isolate the impact of the FM design axes studied in this paper (representation, learning target, trajectory, and dynamics), we control model capacity by using the same denoiser/velocity network across FM and diffusion baselines. Concretely, unless otherwise stated, all models are parameterised by a 4-layer MLP with hidden sizes $[1024, 2048, 2048, 1024]$, following the mixed-type tabular synthesis setup of Zhang et al. (2024). This choice also aligns with the common practice in tabular diffusion, where the denoising network is typically a lightweight MLP (Kotelnikov et al. 2023; Zhang et al. 2024).

For latent-space methods, the encoder and decoder operate on the original mixed-type table and are kept as in the corresponding reference implementations. In particular, TabSyn and TabSynFlow employ Transformer-based VAE components for encoding and decoding, while the diffusion or FM model is trained in the latent space using the same MLP setting. For TabbyFlow, previous work uses transformer + MLP backbones (Guzmán-Cordero et al. 2025). In our benchmark, we additionally report an architecture ablation for TabbyFlow (MLP vs Transformer) to quantify sensitivity to the backbone choice while preserving comparability with MLP-based baselines (Appendix G.3).

Training was conducted with a batch size of 4096 and the ODE was integrated until $t = 1$ using 100 steps [1]. Models were trained for up to 10,000 epochs, with early stopping based on the lowest observed training loss, which is similar to previous studies (Zhang et al. 2024; Guzmán-Cordero et al. 2025), although those studies capped training at 8000 epochs. The integration for synthesis uses the Euler method, including SDE which uses Euler-Maruyama method. All reported numbers are averaged over 20 random seeds of data synthesis. The error bars in the tables denote the $\pm 1$ standard deviation across seeds.

Table 1: Datasets used for the study. $\sum K_d$ is the sum of category counts over categorical variables, while $\max(K_d)$ is the highest cardinality among categorical variables. These statistics summarise both overall categorical complexity and the hardest single-variable regime.

| Abbr. | Dataset Name | #Obs. | #Numerical Variables | #Categorical Variables | $\sum K_d$ | $\max(K_d)$ |
|-------|--------------|-------|----------------------|------------------------|------------|-------------|
| UK | UK Census | 104267 | 1 | 14 | 218 | 74 |
| CA | Canada Census | 32149 | 4 | 21 | 170 | 37 |
| FI | Fiji Census | 84323 | 1 | 18 | 239 | 37 |
| RW | Rwanda Census | 31455 | 1 | 12 | 197 | 37 |
| ID | Indonesia Census | 177429 | 1 | 12 | 62 | 10 |
| AD | Adult | 48842 | 5 | 10 | 122 | 42 |
| CH | Churn Modelling | 10000 | 4 | 7 | 26 | 11 |

---

[1]Unless otherwise specified, this is the default setup

### 4.3 Evaluation Metrics

We evaluate synthetic microdata using two widely adopted criteria in official statistics and statistical disclosure control: **Utility** (Section 4.3.1) and **Disclosure Risk** (Section 4.3.2) (Taub et al. 2020; Little et al. 2022). Our evaluation is motivated by the common practice of tabular microdata, especially in NSOs, for decision-making through interpretable analytical outputs. To support our utility and risk analysis, we also provide additional metrics on data fidelity (Section 4.3.3). Full metric definitions and additional results are provided in Appendix D.

#### 4.3.1 Utility

Utility refers to how well synthetic data supports the same practical analyses as the real data, such as producing descriptive statistics and fitting statistical models. In this paper, we evaluate the utility through two common uses in official statistics: tabulation and inference. Tabulation utility is measured using the ratio-of-counts (ROC) score on univariate and bivariate frequency tables. Inferential utility is measured using confidence interval overlap (CIO) for regression coefficients, a standard task-specific utility metric for synthetic data (Ran et al. 2024; Elliot et al. 2023; Little et al. 2021). We aggregate these three components as:

$$\text{Utility} = \frac{\text{ROC}_{\text{uni}} + \text{ROC}_{\text{biv}} + \text{CIO}}{3}. \tag{4.1}$$

#### 4.3.2 Disclosure Risk

We measure attribute disclosure risk using the targeted correct attribution probability (TCAP) framework for synthetic microdata (Taub et al. 2019; Little et al. 2022). TCAP considers an intruder who (i) knows the individual appearing in the original dataset, (ii) knows the values of that individual for a set of quasi-identifying key variables, and (iii) searches the synthetic data for records matching those keys to infer a sensitive target attribute (Little et al. 2024). For each synthetic record $j$ with key and target values $(K'_j, T'_j)$, we compute the empirical conditional probability in the original data (see Appendix D.2 and Table 8), and aggregate these values across synthetic records. Smaller TCAP indicates a lower probability of correctly inferring the target attribute compared to the baseline and therefore lower attribute disclosure risk.

#### 4.3.3 Additional Metrics

In addition to these primary scores, we also compute widely used diagnostic metrics in tabular deep generative models for data fidelity (Zhang et al. 2024; Guzmán-Cordero et al. 2025; Mueller et al. 2025). Data fidelity is defined as the extent to which synthetic data is statistically indistinguishable from real data, both at the level of marginal distributions and multivariate structure (Adams et al. 2025). This study calculates data fidelity using column-wise Wasserstein distance for continuous attributes, total variation distance for categorical attributes, low-order *shape* and *trend* statistics, a classifier two-sample test (C2ST) detection score, and sample-level $\alpha$-precision and $\beta$-recall.

## 5 Results

In this section, we present and analyse the empirical findings of our study, highlighting the comparative performance of diffusion and flow-matching approaches for tabular data synthesis. As seen from the objective function, all algorithms optimise for utility; disclosure risk is measured post-hoc. We begin by comparing the performance of FM methods with diffusion baselines (Section 5.1). We then examine the number of function evaluations (NFEs), evaluating the utility and risk dynamics between computational efficiency (Section 5.2). Next, we investigate the impact of integration time ($t_{\text{ode}}$) on model performance, with specific attention to the evolution of probability paths (Section 5.3). Finally, we evaluate the effect of incorporating stochastic sampling within the VFM framework (Section 5.4). Collectively, these analyses provide a picture of how modelling choices shape the quality, efficiency, and reliability of synthetic tabular data generation.

Table 2: Comparative Performance of six synthetic data generation algorithms across four common census datasets (UK, CA, FI, RW). We report the mean ± standard deviation across 20 seeds of synthetic data generation. Bold and underline indicate the first and second best performing algorithms for each dataset, respectively. The table emphasises best performance of TabbyFlow in common census benchmarks.

| Evaluation | Algorithm | UK | CA | FI | RW |
|---|---|---|---|---|---|
| Utility | TabDDPM | 0.7823±0.0207 | 0.2819±0.0043 | 0.1266±0.0076 | 0.4125±0.0126 |
| | TabSyn | 0.7617±0.0204 | 0.7366±0.0174 | 0.7085±0.0246 | 0.666±0.0203 |
| | TabSynFlow-OT | 0.7796±0.0188 | 0.7035±0.0129 | 0.6798±0.0132 | 0.6765±0.0221 |
| | TabSynFlow-VP | 0.6696±0.0136 | 0.6403±0.0088 | 0.6014±0.0157 | 0.5875±0.0175 |
| | TabbyFlow-OT | **0.8333±0.0191** | **0.7718±0.0211** | 0.7263±0.0205 | 0.7284±0.0231 |
| | TabbyFlow-VP | 0.8066±0.0197 | 0.7472±0.0188 | **0.7451±0.0216** | **0.7358±0.0185** |
| Risk | TabDDPM | 0.4993±0.0071 | **0.0061±0.0214** | **0.0525±0.0739** | **0.1182±0.1838** |
| | TabSyn | 0.4855±0.0068 | 0.2670±0.0122 | 0.5614±0.0076 | 0.5000±0.0104 |
| | TabSynFlow-OT | 0.4834±0.0085 | 0.2493±0.0149 | 0.5506±0.0065 | 0.4894±0.0107 |
| | TabSynFlow-VP | **0.4630±0.0061** | 0.2073±0.0156 | 0.5208±0.0077 | 0.4655±0.0199 |
| | TabbyFlow-OT | 0.4889±0.0043 | 0.3206±0.0146 | 0.5705±0.0063 | 0.5379±0.0139 |
| | TabbyFlow-VP | 0.4765±0.0072 | 0.3008±0.0148 | 0.5588±0.0074 | 0.5180±0.0105 |

## 5.1 Main Comparison

Tables 2 and 3 report results across seven datasets (five census-type and two standard tabular benchmarks) comparing diffusion baselines (TabDDPM, TabSyn) against flow matching variants in latent space (TabSynFlow-OT/VP) and data space (TabbyFlow-OT/VP). All metrics are reported as mean ± standard deviation across 20 seeds or data synthesis.

Across the four census benchmarks (UK, CA, FI, RW) in Table 2, flow matching is consistently competitive with diffusion, and TabbyFlow achieves the strongest utility overall. The trajectory choice shifts the operating point: TabbyFlow-OT performs best on UK and CA, whereas TabbyFlow-VP performs best on FI and RW, indicating that using VP path can improve utility in some census settings while maintaining low risk. TabSyn remains a strong baseline, and TabSynFlow often matches or improves TabSyn under OT, with VP typically reducing risk at the cost of utility.

The same pattern largely carries over to the three additional datasets (ID, AD, CH) in Table 3. TabbyFlow-OT achieves the highest utility on ID and Adult, while TabSynFlow-OT is strongest on Churn, suggesting that latent-space FM can be preferable when the dataset structure favours compact representation learning. Risk is low across most settings, but remains dataset-dependent: VP generally yields lower risk than OT for a comparable method, consistent with its more conservative sampling behavior. Overall, the results support two practical conclusions: (i) flow matching provides performance comparable to or better than diffusion for mixed-type tabular synthesis, and (ii) the choice between latent-space and data-space FM, as well as OT versus VP trajectories, offers a utility–risk relationship that varies with dataset characteristics.

## 5.2 Number of Function Evaluations between TabSyn and Flow Matching Models

The computational efficiency analysis presented in Figure 1 illustrates the relationship between the number of function evaluations (NFEs) and the performance of synthetic data generation models, highlighting the relationship between computational cost and data quality.

From the figure, it can be seen that the FM models generally converge after approximately 100 NFEs, indicating that using low steps in FM is sufficient. On the other hand, TabbyFlow outperforms TabSyn in all NFE settings based on the utility. Furthermore, TabSyn requires more than 100 steps to outperform FM models, especially TabSynFlow-OT in terms of utility, indicating a higher computational burden, in addition to costly training of VAE+diffusion.

Table 3: Comparative Performance of six synthetic data generation algorithms across three datasets (ID, AD, CH). We report the mean ± standard deviation across 20 seeds of synthetic data generation. Bold and underline indicate the first and second best performing algorithms for each dataset, respectively. The table highlights the TabbyFlow and TabSynFlow-OT which stands out as a strong alternative to TabSyn.

| Evaluation | Algorithm | ID | AD | CH |
|---|---|---|---|---|
| Utility (↑) | TabDDPM | 0.7993±0.0108 | 0.6676±0.0092 | 0.8227±0.0130 |
| | TabSyn | 0.8707±0.0264 | 0.7596±0.0139 | 0.8663±0.0124 |
| | TabSynFlow-OT | 0.8981±0.0300 | 0.7560±0.0139 | **0.8784±0.0184** |
| | TabSynFlow-VP | 0.7814±0.0100 | 0.6843±0.0100 | 0.8126±0.0149 |
| | TabbyFlow-OT | **0.9191±0.0139** | **0.7720±0.0231** | 0.7966±0.0098 |
| | TabbyFlow-VP | 0.8473±0.0372 | 0.7581±0.0165 | 0.8066±0.0102 |
| Risk (↓) | TabDDPM | 0.6493±0.0116 | 0.4369±0.0134 | 0.0718±0.0610 |
| | TabSyn | 0.6624±0.0122 | 0.4805±0.0207 | 0.0630±0.0625 |
| | TabSynFlow-OT | 0.6616±0.0119 | 0.4693±0.0149 | 0.0429±0.0438 |
| | TabSynFlow-VP | 0.6464±0.0140 | **0.4131±0.0199** | **0.0246±0.0345** |
| | TabbyFlow-OT | 0.6556±0.0133 | 0.5443±0.0177 | 0.0773±0.0579 |
| | TabbyFlow-VP | **0.6447±0.0096** | 0.5037±0.0172 | 0.0753±0.0708 |

In low-NFE settings, where flow-based models are designed to operate efficiently, TabSyn performed poorly. TabbyFlow-VP, whilst yielding lower utility, also achieves a lower disclosure risk and still outperforms TabSyn when NFEs are ≤ 32. Meanwhile, TabSynFlow has synthetic data with the lowest disclosure risk, while also having higher utility than TabSyn when NFEs are ≤ 16. This early-stage advantage positions flow matching models as promising candidates for applications operating under strict computational constraints. In summary, while TabSyn can achieve competitive utility, it does so at the cost of significantly higher NFEs. Flow-matching models offer a more efficient alternative, especially in scenarios where computational efficiency is critical.

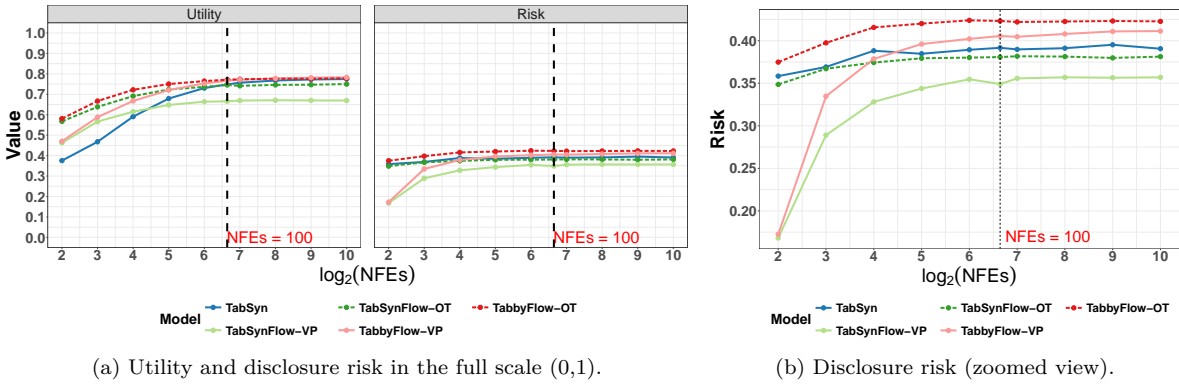

(a) Utility and disclosure risk in the full scale (0,1).    (b) Disclosure risk (zoomed view).

Figure 1: **Average utility (↑) and disclosure risk (↓) as a function of the number of function evaluations (NFEs).** Results are averaged across four datasets. **(a)** Overall trends shown on the full scale $[0, 1]$. **(b)** Zoomed view of the disclosure risk to highlight differences between methods. TabSyn (blue, solid) is compared against flow-matching models (TabSynFlow-OT, TabSynFlow-VP, TabbyFlow-OT, TabbyFlow-VP). For flow-matching models, colours and line styles indicate the probability path (VP: lighter colours and solid lines; OT: darker colours and dashed lines). The vertical dashed line marks 100 NFEs, corresponding to the default implementation. Across datasets, flow-matching models saturate after approximately 100 NFEs, achieving competitive utility while requiring substantially fewer function evaluations than TabSyn.

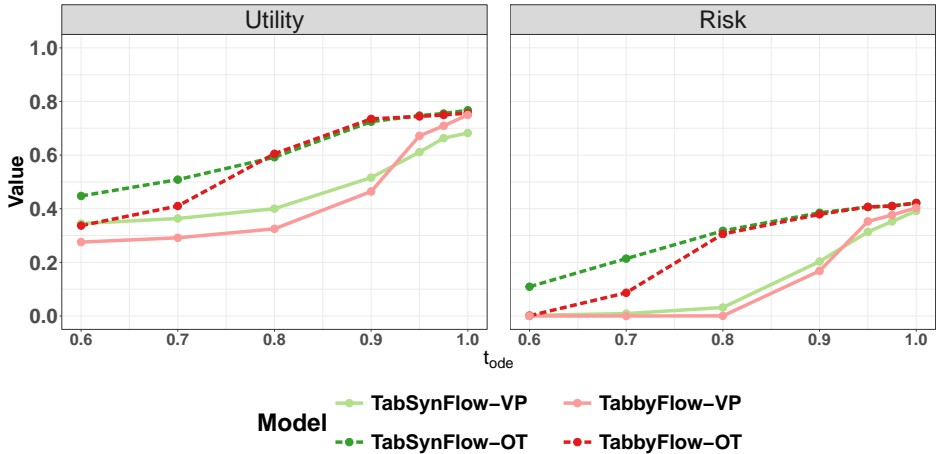

Figure 2: **Average utility (↑) and risk (↓) against ODE integration time** ($t_{\text{ode}}$) for **TabSynFlow** (green) and **TabbyFlow** (red) using OT and VP paths. Colours and line styles indicate method and path (VP light and solid, OT dark and dashed). OT paths enable early stopping without significant utility loss, whereas VP paths require full integration to achieve competitive performance.

### 5.3   Flow Matching Results based on Integration Time

In standard flow matching implementations, the ODE is fully integrated; however, early termination is possible in practice. Figure 2 provides an analysis of the effect of ODE integration times ($t_{\text{ode}}$) on utility and risk, averaged across all datasets. For TabSynFlow-OT, the utility remains consistently high throughout integration times, demonstrating robust performance even with early stopping. As in $t_{\text{ode}} \to 1.0$, the performance across all methods tends to plateau.

In contrast, both TabSynFlow and TabbyFlow using the VP path (solid and lighter lines) exhibit lower initial utility at $t_{\text{ode}} = 0.6$ and increase substantially when $t_{\text{ode}} \geq 0.9$, suggesting that VP introduces disruptive noise early in the process and require full integration to achieve adequate synthetic data quality. From these results, we can see that the consistent superiority of OT paths across integration times highlights their robustness where early termination is desirable for computationally constrained applications. The dataset-specific results provided in Appendix E.3, including when $0.9 \leq t_{\text{ode}} \leq 1$, highlighting the importance of considering the characteristics of the dataset in the selection of integration parameters.

### 5.4   SDE Implementation of Variational Flow Matching

In this subsection, we investigate whether stochastic differential equation (SDE) sampling yields results comparable to ODE sampling. Tables 4 and 5 present a comparative analysis of the ODE and SDE formulations, with additional results provided in Appendix F.5 for alternative diffusion coefficients.

In general, it can be seen that the SDE approach performs as expected theoretically, producing marginal distributions similar to those generated by ODE  Eijkelboom et al. (2024) while, in some cases, producing superior performance. This is evidenced by the close alignment of the utility and risk values in both formulations. This empirical validation confirms the mathematical foundation presented in the paper.

Beyond theoretical equivalence, SDE sampling offers a practical advantage by improving the ODE results in some cases. For example, for Fiji in both configurations and for Adult and Churn under VP, SDE-based VFM yields higher-utility synthetic data with lower risk, a dual gain that is uncommon in practice, where improvements usually appear in only one of the two metrics.

Table 4: Comparison of average utility and risk between ODE and SDE performance for TabbyFlow-OT. $g_t$ used in the SDE is VP-based $\sigma_t$.

| Dataset | ODE | | SDE | |
|---|---|---|---|---|
| | Utility | Risk | Utility | Risk |
| UK | 0.8333 | 0.4889 | 0.8289 | 0.4889 |
| Canada | 0.7718 | 0.3206 | 0.7778 | 0.3213 |
| Fiji | 0.7263 | 0.5705 | 0.7330 | 0.5688 |
| Rwanda | 0.7284 | 0.5379 | 0.7426 | 0.5382 |
| Indonesia | 0.9191 | 0.6556 | 0.9049 | 0.6552 |
| Adult | 0.7720 | 0.5443 | 0.7747 | 0.5483 |
| Churn | 0.7966 | 0.0773 | 0.7987 | 0.0843 |

Table 5: Comparison of average utility and risk between ODE and SDE performance for TabbyFlow-VP. $g_t$ used in the SDE is OT-based $\sigma_t$.

| Dataset | ODE | | SDE | |
|---|---|---|---|---|
| | Utility | Risk | Utility | Risk |
| UK | 0.8066 | 0.4765 | 0.8031 | 0.4761 |
| Canada | 0.7472 | 0.3008 | 0.7487 | 0.3037 |
| Fiji | 0.7451 | 0.5588 | 0.7489 | 0.5576 |
| Rwanda | 0.7358 | 0.5180 | 0.7461 | 0.5192 |
| Indonesia | 0.8473 | 0.6447 | 0.8608 | 0.6457 |
| Adult | 0.7581 | 0.5037 | 0.7625 | 0.5002 |
| Churn | 0.8066 | 0.0753 | 0.8067 | 0.0651 |

## 6 Discussion and Concluding Remarks

### 6.1 Discussion

The experimental results demonstrate the performance characteristics of flow-based deep generative models for tabular data synthesis, highlighting how utility and disclosure risk vary across algorithms and datasets.

**Performance Comparison (Section 5.1).** Flow matching models consistently outperform TabDDPM on most datasets, with TabbyFlow generating synthetic data of the highest utility in nearly all cases. This increased utility is generally accompanied by an elevated disclosure risk, reflecting a positive empirical correlation between utility and risk. An interesting finding is that using a VP path in TabbyFlow can yield higher utility with lower disclosure risk, even though VP paths are not commonly used in FM studies. TabSyn remains a reliable benchmark, while TabSynFlow-OT shows targeted improvements over TabSyn in selected cases but not universally. These results highlight flow matching, particularly TabbyFlow, as a viable alternative for tabular data synthesis, with trajectory choice giving different performance in terms of data utility and disclosure risk.

**Efficiency via NFEs (Section 5.2).** Flow matching models, particularly TabbyFlow-OT and TabSynFlow-OT, demonstrate strong utility with a relatively small number of function evaluations (NFEs), typically between 64 and 128, whereas TabSyn requires substantially more NFEs to reach comparable performance. In particular, TabbyFlow can achieve near-optimal utility within NFEs 128, and even at very low budgets (four evaluation steps), meaningful results are observed. The actual impact of NFEs, however, depends heavily on data size, feature complexity, and model architecture. For instance, generating small datasets like CH is fast, but larger datasets such as UK or ID require more time and batch-wise processing to avoid out-of-memory errors, particularly on personal computers. Overall, lower NFEs values generally reflect greater computational efficiency, making flow-based models especially attractive for resource-constrained environments.

**Integration Time and Early Stopping (Section 5.3).** OT paths outperform VP paths in both TabSynFlow and TabbyFlow, maintaining high utility earlier in the generation process. VP paths require full integration to achieve competitive performance. However, both trajectories are prone to reduced utility and / or increased risk when the integration time approaches one ($t_{\text{ode}} \to 1$). Practical strategies to mitigate this include early stopping ($0.9 \le t_{\text{ode}} < 1.0$) or velocity clipping[2]. In general, OT trajectories provide a more stable and reliable approach to tabular synthesis.

**Deterministic vs. Stochastic Sampling (Section 5.4).** The application of SDE solvers within the VFM framework presents a more complex picture than initially hypothesised. Although theoretically established that SDE recovers the same marginal distributions as ODE, our empirical results reveal numerical differences. In particular, on datasets such as Fiji, SDE sampling led to a simultaneous increase in utility and a reduction in risk. This dual improvement suggests that the controlled stochasticity of the SDE can act as a beneficial

---

[2]e.g., `clip(1-1/(1-t), 0, m)`, m is the maximum value allowed

regulariser or enable more effective exploration of the data space, rather than merely introducing destructive noise. However, since performance gains were not observed across all datasets, the utility of SDEs appears to be data-dependent. Thus, the choice between deterministic and stochastic samplers is not about universal dominance but about suitability to a specific dataset.

## 6.2 Concluding Remarks

This study has presented a comprehensive evaluation of flow matching models for tabular data synthesis, comparing them with state-of-the-art diffusion-based approaches. Flow matching, particularly TabbyFlow, achieves higher utility across datasets, while disclosure risk follows patterns determined by the empirical relationship between model utility and privacy, rather than optimisation for risk reduction. OT trajectories provide a strong default with stable utility and low sensitivity to integration time, whereas VP typically lowers risk at some utility cost, so the choice should depend on application requirements.

TabbyFlow is also computationally efficient, reaching competitive performance with fewer function evaluations. We recommend OT with earlier integration time ($0.9 \leq t_{\text{ode}} < 1.0$) and 64–128 NFEs as a practical trade-off between quality and computational cost, with 4–8 NFEs remaining usable in resource-constrained settings. SDE sampling can sometimes improve both utility and risk, but its benefits are dataset-dependent, so it is best treated as an optional extension validated per dataset.

Overall, both latent-space and data-space flow matching (including variational formulations) are effective for tabular synthesis in our study. Future work should expand evaluation beyond census-style tables to higher-dimensional domains (for example, health or genomics), study mixed continuous–categorical generation more directly (including hybrid conditional and discrete FM (Gat et al. 2024)), and incorporate formal privacy protection such as differential privacy or federated learning (Little et al. 2023) for sensitive deployments.

### Broader Impact Statement

Synthetic tabular microdata can help statistical agencies and other data custodians broaden access to data for research and policy analysis when releasing unit records is restricted. However, disclosure risk can persist through linkage or attribute inference, and low-fidelity synthesis can bias results, especially for small subpopulations or rare categories. This work provides empirical evidence on how flow-based design and sampling choices shift the utility–risk relation under evaluation practices used in official statistics. Any deployment should be dataset-specific and include domain review, clear communication of intended use, and risk assessment reported alongside utility. These considerations also apply to other sensitive tabular domains such as health, finance, and public services.

### Acknowledgments

B.I. Nasution is supported by the Indonesia Endowment Fund for Education Agency (LPDP), and his research visit to the University of Amsterdam was supported by the Turing Scheme Programme.

### LLM Usage Statement.

During the preparation of this manuscript, the author used large language models (LLMs) such as ChatGPT and Perplexity primarily for editing and improving the readability of the text. All analyses and claims are entirely the author's original work and responsibility.

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

## Supplementary Materials

## A   Related Work

**Diffusion Models for Tabular Data.** Denoising diffusion probabilistic models (DDPMs) have shown great success across domains like vision and language, but their application to tabular data poses unique challenges due to feature heterogeneity and small dataset sizes. TabDDPM (Kotelnikov et al. 2023) addresses this through a dual process: Gaussian diffusion for continuous variables and multinomial diffusion for categorical features. It achieves state-of-the-art (SOTA) utility and privacy performance, outperforming the GAN and VAE baselines. However, its computational cost remains high because of iterative sampling and separate handling of feature types, which may weaken correlation modelling. TabSyn (Zhang et al. 2024) employs a transformer-VAE architecture followed by a diffusion model, while the continuous diffusion model for mixed type tabular data (CDTD) (Mueller et al. 2025) is a score-matching diffusion model that noises continuous values and categorical embeddings and samples via ODE integration. These two methods improve the utility of tabular data, particularly compared to TabDDPM and other recent SOTAs.

**FM and its Variational Extension.** Flow matching (FM) (Lipman et al. 2023; Liu et al. 2022; Albergo et al. 2023) offers an alternative to diffusion by learning the deterministic dynamics of probability paths. FM has also been developed in terms of latent models for image generation (Dao et al. 2023; Schusterbauer et al. 2024). In addition, VFM (Eijkelboom et al. 2024) reformulates FM as a variational inference problem, allowing flow matching through a variational distribution. Consequently, VFM can be implemented to mixed-type data, such as tabular data, through mean-field variational inference. On the other hand, (Guzmán-Cordero et al. 2025) extended the VFM specific for the exponential family that also supports mixed-type variables through moment matching and implemented it in tabular data, called TabbyFlow. In addition to tabular, VFM has also been implemented for image generation (Matişan et al. 2025), controlled molecular generation (Eijkelboom et al. 2025), non-Euclidean geometry (Zaghen et al. 2025a;b), and climate modeling (Finn et al. 2025).

**FM Trajectory and Interpolant.** Trajectory design is of paramount importance in the field of FM. The optimal transport (OT) path is widely preferred because of their straightforward linear interpolation. In contrast, the variance-preserving (VP) path, which takes inspiration from diffusion models, is somewhat underrated (Lipman et al. 2023; Liu et al. 2022). Recent studies also introduced different interpolants such as cosine (Albergo & Vanden-Eijnden 2023), and logit-normal (Black Forest Labs 2025). Although the theoretical framework of FM accommodates all types of path, there is a notable gap in empirical evaluation concerning the practical application of different paths, particularly in the realm of tabular data. To address this gap, we present the potential of different paths for tabular data synthesis, despite the fact that OT is the primary implementation.

## B   TabSyn: Latent Diffusion for Tabular Data

TabSyn (Zhang et al. 2024) represents a state-of-the-art generative approach for mixed-type tabular data. It integrates a transformer-based Variational Autoencoder (VAE) with a score-based diffusion model operating in latent space. The design of TabSyn addresses several key challenges in tabular data synthesis including heterogeneous feature types, complex inter-feature dependencies, and the inefficiency of diffusion processes when employed on high-dimensional structured data.

The TabSyn framework comprises two stages. First, a **VAE** maps the raw tabular data into a continuous latent space. Second, a **diffusion model** is trained to capture the underlying distribution with this latent space. To handle different data types, TabSyn uses column-wise tokenisers and detokenisers, transformer-based encoders/decoders, and an adaptive $\beta$-VAE objective. The encoder learns structured latent representations, while the decoder ensures accurate and type-consistent data reconstruction.

Within the latent space, noise is introduced through a forward process defined as: $z_t = z_0 + \sigma(t)\varepsilon$. Given linear noise schedule $t$, the corresponding reverse process is an SDE:

$$\mathrm{d}z_t = -2t\nabla_z \log p_t(z_t)\,\mathrm{d}t + \sqrt{2t}\,\mathrm{d}\omega_t. \tag{B.1}$$

where $\omega_t$ denotes a Wiener process. The model is trained using denoising score matching, with a neural network $\varepsilon_\theta(z_t, t)$ that estimates the scoring function $\nabla_{z_t} \log p_t(z_t) = -\varepsilon_\theta(z_t, t)/t$ under linear noise schedule.

TabSyn surpasses traditional GAN and VAE based models as well as the latest diffusion models like TabD-DPM and STaSy, across a variety of datasets and evaluation metrics - including unvariate distribution and pairwise correlation preservation, and downstream ML performance. Furthermore, its latent-space formulation allows for missing value imputation via latent inpainting, achieving competitive results in terms of data quality, diversity, and privacy preservation.

## C  Algorithms

### C.1  TabSynFlow

---

**Algorithm 1** An iteration of TabSynFlow training (flow matching part)

---

**Input:** Latent representation from VAE $\mathcal{Z}_1$, time function for mean $\alpha_t$, time function for variance $\sigma_t$, conditional velocity field NN(.)

---

1: $z_1 \sim p_{\mathcal{Z}_1}$
2: $z_0 \sim \mathcal{N}(0, I)$
3: $t \sim \text{Uniform}(0, 1)$
4: $z_t = \alpha_t z_1 + \sigma_t z_0$
5: Calculate $u_t(z_t | z_1)$ using equation 2.4
6: Predict $v_t^\theta(z_t) = \text{NN}(z_t, t)$
7: Calculate $\mathcal{L}_{\text{TabSynFlow}}$

---

**Algorithm 2** TabSynFlow sample generation

---

**Input:** Trained conditional velocity field NN(.), discrete step interval $\Delta$, trained decoder from VAE Dec(.)

---

1: $\tilde{z}_0 \sim \mathcal{N}(0, I)$.
2: $t = 0$
3: **while** $t \leq 1$ **do**
4:      $\tilde{z}_{t+\Delta} = \tilde{z}_t + \Delta \cdot \text{NN}(\tilde{z}_t, t)$
5:      $t {+}{=} \Delta$
6: **end while**
7: $\tilde{x} = \text{Dec}(\tilde{z}_1)$
8: $\tilde{\mathcal{D}} = \text{InverseTransform}(\tilde{x})$
**Return** $\tilde{\mathcal{D}}$

---

### C.2  TabbyFlow

---

**Algorithm 3** TabbyFlow training iteration

---

**Input:** Preprocessed data, Function of time for mean $\alpha_t$, function of time for variance $\sigma_t$, variational parameter model NN(.)

---

1: $x_1 \sim p_{data}$
2: $x_0 \sim \mathcal{N}(0, I)$
3: $t \sim \text{Uniform}(0, 1)$
4: $x_t = \alpha_t x_1 + \sigma_t x_0$
5: Predict $\theta_t(x_t) = \text{NN}(x_t, t)$
6: Calculate $\mathcal{L}_{\text{TabbyFlow}}(\theta)$

---

**Algorithm 4** TabbyFlow sample generation

---

**Input:** Time function for mean $\alpha_t$, time function for variance $\sigma_t$, variational parameter model NN(.), discrete step interval $\Delta$, noise schedule $g_t$ (for SDE)

---

1: $\tilde{x}_0 \sim \mathcal{N}(0, I)$
2: $t = 0$
3: **while** $t \leq 1$ **do**
4:      $\tilde{\theta}_t(\tilde{x}_t) = \text{NN}(\tilde{x}_t, t)$
5:      Calculate $v_t^\theta(\tilde{x}_t)$ using equation 2.7
    **ODE:**
6:      $\tilde{x}_{t+\Delta} = \tilde{x}_t + \Delta \cdot v_t^\theta(\tilde{x}_t)$
    **SDE:**
7:      Calculate $s_t^\theta(x)$ using equation 2.9
8:      $\tilde{x}_{t+\Delta} = \tilde{x}_t + \Delta \left[ v_t^\theta(\tilde{x}_t) + \frac{g_t^2}{2} s_t^\theta(\tilde{x}_t) \right] + g_t \sqrt{\Delta} \mathcal{N}(0, I)$
9:      $t {+}{=} \Delta$
10: **end while**
11: $\tilde{\mathcal{D}} = \text{InverseTransform}(\tilde{x}_1)$
**Return** $\tilde{\mathcal{D}}$

---

## D  Evaluation Details

This section provides detailed information on the evaluation metrics and implementation procedures used to assess the quality and privacy aspects of the synthetic tabular data generated in our experiments. This additional context aims to ensuring transparency and reproducibility, allowing readers to fully understand

how each score is derived and how our experimental protocol aligns with established practices in tabular data synthesis evaluation.

### D.1 Utility Metrics

Utility measures the extent to which synthetic data faithfully reproduces the statistical relationships and analytical properties of the original data. We employ two complementary utility metrics:

#### D.1.1 Ratio of Counts (ROC)

ROC evaluates the fidelity of frequency tables and cross-tabulations by comparing cell counts between original and synthetic data (Nasution et al. 2026). For each cell, the ROC is:

$$\text{ROC}_{\text{cell}} = \frac{\min(\mathcal{Y}_{count}, \mathcal{Y}'_{count})}{\max(\mathcal{Y}_{count}, \mathcal{Y}'_{count})}, \tag{D.1}$$

where $\mathcal{Y}_{count}$ and $\mathcal{Y}'_{count}$ denote cell counts in the original and synthetic data respectively. A value of 1 indicates perfect agreement; 0 indicates extreme disagreement.

ROC is computed across two types of tabulations:

- **Univariate**: Frequency distribution of each key variable independently

- **Bivariate**: Cross-tabulation of all pairs of key variables

The final $\text{ROC}_{\text{uni}}$ score is the mean across all univariate tables, and $\text{ROC}_{\text{biv}}$ is the mean across all bivariate tables, i.e. cross-tabulation.

#### D.1.2 Confidence Interval Overlap (CIO)

CIO assesses how well statistical inference (specifically logistic regression) conducted on synthetic data recovers the confidence intervals from the original data. For each regression coefficient, CIO calculates the normalized overlap of 95% confidence intervals:

$$\text{CIO}_k = 0.5 \left[ \frac{\text{overlap}}{u - l} + \frac{\text{overlap}}{u' - l'} \right], \tag{D.2}$$

where $[l, u]$ and $[l', u']$ are the confidence intervals of regression coefficients from original and synthetic models respectively, and overlap $= \min(u, u') - \max(l, l')$. A CIO of 1 indicates identical intervals, whereas CIO $<$ 0 indicates no overlap, therefore it is truncated to zero (Elliot et al. 2023). The final CIO score is the mean overlap across all regression coefficients and all fitted models.

#### D.1.3 Aggregate Utility Score

The final utility score for each dataset is computed as:

$$\text{Utility} = \frac{\text{ROC}_{\text{uni}} + \text{ROC}_{\text{biv}} + \text{CIO}}{3} \tag{D.3}$$

For each dataset, we compute ROC across all univariate and bivariate frequency tables, and CIO for regression models predicting a defined target variable. Mainly, we used logistic regression, except for target variable CreditScore and EstimatedSalary in Churn data that uses linear regression since they are continuous variables. The pseudocode to compute ROC and CIO can be seen in Algorithms 5 and 6.

---

**Algorithm 5** ROC calculation

---

**Input:** Original data $\mathcal{D}$, synthetic data $\tilde{\mathcal{D}}$, key variables $K$

1: **for** each variable $k$ in $K$ **do**
2:      **for** each category $c$ of $k$ **do**
3:          Compute $\mathrm{ROC_{uni}}[c] = \frac{\min(\mathcal{Y}_{count}(c), \mathcal{Y}'_{count}(c))}{\max(\mathcal{Y}_{count}(c), \mathcal{Y}'_{count}(c))}$
4:      **end for**
5: **end for**
6: $\mathrm{ROC_{uni}} \leftarrow$ mean of $\mathrm{ROC}_{uni}[c]$ over all variables and categories
7:
8: **for** each pair $(k_1, k_2)$ in $K$ **do**
9:      **for** each cell $(c_1, c_2)$ **do**
10:          Compute $\mathrm{ROC_{biv}}[c_1, c_2] = \frac{\min(\mathcal{Y}_{count}(c_1,c_2), \mathcal{Y}'_{count}(c_1,c_2))}{\max(\mathcal{Y}_{count}(c_1,c_2), \mathcal{Y}'_{count}(c_1,c_2))}$
11:      **end for**
12: **end for**
13: $\mathrm{ROC_{biv}} \leftarrow$ mean of $\mathrm{ROC_{biv}}[c_1, c_2]$ over all pairs and cells
14: **Return** $\mathrm{ROC_{uni}}$, $\mathrm{ROC_{biv}}$

---

**Algorithm 6** CIO calculation

---

**Input:** Original data $\mathcal{D}$, synthetic data $\tilde{\mathcal{D}}$, key variables $K$, target variable $T$

1: Fit regression model (predictors $K$, target $T$) on $\mathcal{D}$
2: Fit regression model (predictors $K$, target $T$) on $\tilde{\mathcal{D}}$
3: **for** each coefficient $i$ **do**
4:      Compute $[l, u]$ from model on $\mathcal{D}$
5:      Compute $[l', u']$ from model on $\tilde{\mathcal{D}}$
6:      overlap $= \max(0, \min(u, u') - \max(l, l'))$
7:      $\mathrm{CIO}_i = 0.5 \left[ \frac{\text{overlap}}{u-l} + \frac{\text{overlap}}{u'-l'} \right]$
8: **end for**
9: $\mathrm{CIO} \leftarrow$ mean of $\mathrm{CIO}_i$ over all coefficients
10: **Return** CIO

---

## D.2 Disclosure Risk Metric

We measure disclosure risk as the probability that an adversary, possessing the quasi-identifier (key variables, $K$), can correctly guess the sensitive value (target $T$) for a record by matching between the synthetic and original data. This approach, adapted from Taub et al. (2019), quantifies the additional privacy risk introduced by synthetic data at a per-record granularity, adjusting for the baseline (based on a random draw from $T$), defined by the distribution of $T$ within each equivalence class (combination of $K$) in the original data.

For each synthetic record, we first identify the group of records in the synthetic data that share the same values for the key variables, as well as the more specific group that also matches on the target variable. The proportion is then calculated as the number of records sharing both key and target values divided by the number sharing only the key values. This proportion is used to filter which synthetic records are included in the TCAP calculation: only those in groups where the proportion meets or exceeds a specified threshold (represented by the parameter $\tau$ - which represents an assumed lower bound on the acceptable level of uncertainty for the adversary) are considered, while records in less concentrated groups are excluded. This filtering is applied solely within the synthetic data; the actual TCAP risk for each retained record is subsequently calculated based on the target distribution in the original data for the corresponding key group.

**Targeted Correct Attribution Probability** ($TCAP$) estimates the probability that the adversary can match the synthetic target $T'_j$ using the equivalence class from the original data:

$$\mathrm{TCAP}'_j = \mathbb{P}_{\mathcal{D}}(T'_j | K'_j) = \frac{\sum_{i=1}^{n} \mathbb{I}[T_i = T'_j, \, K_i = K'_j]}{\sum_{i=1}^{n} \mathbb{I}[K_i = K'_j]} \tag{D.4}$$

Intuitively, $\mathrm{TCAP}'_j$ shows the chance that target $T'_j$ could arise in the real data for key $K'_j$—that is, it reflects the "true match" risk based on empirical frequencies in the original dataset.

**Within-Equivalence Class Attribute Probability** (*WEAP*) serves as a baseline: if the adversary could only guess randomly within the equivalence class $K_j'$, their success probability would be

$$\text{WEAP}_j = \mathbb{P}_{\mathcal{D}}(T_j|K_j) = \frac{\sum_{i=1}^{n} \mathbb{I}[T_i = T_j, K_i = K_j]}{\sum_{i=1}^{n} \mathbb{I}[K_i = K_j]} \tag{D.5}$$

This is identical in calculation to $TCAP_j'$, but conceptually $WEAP_j$ captures the average success rate without access to synthetic information—i.e., a "random guess" adversary.

**Per-record disclosure risk** is then the normalised improvement of adversarial success above this baseline:

$$\text{Risk}_j = \max \left\{ 0, \frac{\text{TCAP}_j' - \text{WEAP}_j}{1 - \text{WEAP}_j} \right\} \tag{D.6}$$

A value near 0 indicates synthetic data provides no additional risk beyond baseline, while a value near 1 signals a maximum increase in disclosure risk. Negative values (adversary performs worse than random) are truncated to zero by convention (Ran et al. 2024; Elliot et al. 2023).

*Intuition:* This metric isolates the extra "leakage" in matching risk—beyond what already exists in the univariate distribution of the target — arising from access to the synthetic data. It thus accounts for singling out, linkage and inference risks (due to the empirical distribution of sensitive values per equivalence class) identified by Article 29 Data Protection Working Party (2025) as being critical determinants of risks to anonymity. The disclosure risk for the dataset is the mean of $R_j$ over all synthetic records meeting the $\tau$ threshold. The pseudocode to calculate TCAP can be seen in Algorithm 7

---

**Algorithm 7** TCAP calculation

---

**Input:** Original data $\mathcal{D}$, synthetic data $\tilde{\mathcal{D}}$, key variables $K$, target variable $T$, threshold $\tau$

1: **for** each synthetic record $j$ in $\tilde{\mathcal{D}}$ **do**
2:     Identify equivalence class $K_j'$ and $[T_j', K_j']$ in synthetic data
3:     $p' \leftarrow \frac{\mathbb{I}[T_j', K_j']}{\mathbb{I}[K_j']}$
4:     **if** $p' < \tau$ **then**
5:         Continue to next $j$                                ▷ Skip if class too small
6:     **end if**
7:     Identify all records in $\mathcal{D}$ with $K_i = K_j'$, denote as class $C$
8:     **if** $C$ is empty **then**
9:         $R_j \leftarrow 0$, go to next $j$
10:     **end if**
11:     Calculate $\text{TCAP}_j'$ using equation D.4
12:     Calculate $\text{WEAP}_j$ using equation D.5
13:     Calculate $\text{Risk}_j$ using equation D.6
14: **end for**
15: Risk $\leftarrow$ mean of all $R_j$

---

### D.3 Additional Diagnostic Metrics

To complement Utility and Disclosure Risk, we report standard diagnostic metrics that quantify low-order distributional fidelity, detectability, and sample-wise quality. These diagnostics are provided for completeness and to facilitate comparison with prior synthetic tabular data benchmarks. Unless stated otherwise, lower values indicate closer agreement between the real and synthetic distributions.

#### D.3.1 Discrepancy between Synthetic and Real Data Distributions

We measure the discrepancy between the synthetic and real distributions using the Wasserstein distance (WD) and Maximum Mean Discrepancy (MMD). In particular, mixed-type tabular records are first mapped

into a common vector space via the same preprocessing applied by the data loader (continuous normalisation and categorical encoding), yielding vectors $x \in \mathbb{R}^d$. On large datasets, computing the full distance is expensive due to pairwise computations between samples. Study conducted by Jolicoeur-Martineau et al. (2024) also evaluated using these measure on data with less than 5,000 rows. We therefore use a batched evaluation protocol: we draw $K$ non-overlapping (or randomly sampled) batches of size $n = 5,000$ from the real and synthetic datasets and report the average distance as seen in equation D.8.

$$\mathrm{WD}(P_r, P_g) = \frac{1}{M} \sum_{k=1}^{M} W\left(\mathcal{B}^{(k)}, \tilde{\mathcal{B}}^{(k)}\right), \tag{D.7}$$

$$\mathrm{MMD}^2(P_r, P_g) = \mathbb{E}_{x,x' \sim P_r}[k(x, x')] + \mathbb{E}_{y,y' \sim P_g}[k(y, y')] - 2\,\mathbb{E}_{x \sim P_r, y \sim P_g}[k(x, y)]. \tag{D.8}$$

where $\mathcal{B}^{(k)}$ and $\tilde{\mathcal{B}}^{(k)}$ are the $k$-th batches of the preprocessed real and synthetic samples, respectively. This provides a practical Monte Carlo estimate of the distributional discrepancy under limited compute.

### D.3.2 Shape score (column-wise density discrepancy)

We summarisecolumn-wise density discrepancies using a *shape* score computed as the average of (i) the Kolmogorov–Smirnov statistic (KST) over continuous columns and (ii) TVD over categorical columns. For $j$-th continuous variable:

$$\mathrm{KST}_j = \sup_x \left| F_j(x) - F_j'(x) \right|. \tag{D.9}$$

For $k$-th categorical variable, we use $\mathrm{TVD}(p_k, p_k')$ from equation D.10.

$$\mathrm{TVD}(p_k, p_k') = \frac{1}{2} \sum_{k=1}^{D_{\mathrm{cat}}} |p_k(\omega) - p_k'(\omega)|. \tag{D.10}$$

The aggregate shape score is

$$\mathrm{Shape} = \frac{1}{D_{\mathrm{num}} + D_{\mathrm{cat}}} \left( \sum_{j=1}^{D_{\mathrm{num}}} \mathrm{KST}_j + \sum_{k=1}^{D_{\mathrm{cat}}} \mathrm{TVD}(p_k, p_k') \right). \tag{D.11}$$

### D.3.3 Trend score (pairwise dependence discrepancy)

We quantify how well pairwise dependence is preserved using a *trend* score that combines (i) Pearson correlation discrepancies for continuous–continuous pairs and (ii) contingency table discrepancies for categorical–categorical pairs.

**Continuous–continuous (Pearson).** For continuous columns $a, b \in \mathcal{D}_{\mathrm{num}}$, let $\rho_{a,b}^{\mathcal{D}}$ and $\rho_{a,b}^{\tilde{\mathcal{D}}}$ be Pearson correlations computed on $\mathcal{D}$ and $\tilde{\mathcal{D}}$. Since $\rho \in [-1, 1]$, we scale absolute differences by $1/2$ so the score lies in $[0, 1]$:

$$\mathrm{Trend}_{\mathrm{cont}} = \frac{1}{2|\mathcal{P}_{\mathcal{D}_{\mathrm{num}}}|} \sum_{(a,b) \in \mathcal{P}_{\mathcal{D}_{\mathrm{num}}}} \left| \rho_{a,b}^{\mathcal{D}} - \rho_{a,b}^{\tilde{\mathcal{D}}} \right|, \tag{D.12}$$

where $\mathcal{P}_{\mathcal{D}_{\mathrm{num}}}$ is the set of unordered continuous pairs.

**Categorical–categorical (contingency similarity).** For categorical columns $u, v \in \mathcal{D}_{\mathrm{cat}}$, let $R_{u,v}(\alpha, \beta)$ and $S_{u,v}(\alpha, \beta)$ denote the empirical joint frequencies in the contingency tables on $\mathcal{D}$ and $\tilde{\mathcal{D}}$. We compute a TVD-style discrepancy:

$$\mathrm{Trend}_{\mathrm{cat}} = \frac{1}{2|\mathcal{P}_{\mathcal{D}_{\mathrm{cat}}}|} \sum_{(u,v) \in \mathcal{P}_{\mathcal{D}_{\mathrm{cat}}}} \sum_{\alpha \in \Omega_u} \sum_{\beta \in \Omega_v} |R_{u,v}(\alpha, \beta) - S_{u,v}(\alpha, \beta)|, \tag{D.13}$$

where $\mathcal{P}_{\mathcal{D}_{\mathrm{cat}}}$ is the set of unordered categorical pairs.

When both continuous and categorical pairs are present, we report overall trend score as their average.

### D.3.4 Detection score via classifier two-sample test (C2ST)

We measure detectability by training a binary classifier to distinguish real from synthetic records. We construct a labeled dataset by assigning label 1 to samples from $\mathcal{D}$ and label 0 to samples from $\tilde{\mathcal{D}}$, then evaluate the classifier on a held-out split. The C2ST score is the test accuracy $\text{Acc} \in [0, 1]$, where values close to 0.5 indicate low detectability.

### D.3.5 Sample-wise fidelity and coverage: $\alpha$-precision and $\beta$-recall

We report $\alpha$-precision and $\beta$-recall, which summarise sample-wise fidelity and coverage of the real distribution. Informally, $\alpha$-precision is the fraction of synthetic samples that fall within the "typical" (high-density) region of the real distribution, while $\beta$-recall is the fraction of real samples covered by the typical region of the synthetic distribution. We compute these scores using the procedure in, based on minimum-volume (high-density) sets estimated from samples.

### D.4 Reproducibility

This appendix provides the procedural summary sufficient for manual replication or verification by independent researchers. For computational resource, we run all experiments using Python 3.11 with library torch 2.3.0 in an L40S GPU. However, since the model used in this study is an MLP, it is also possible to reproduce this experiment under regular GPU like GTX/RTX family. Table 6- 8 showed detailed variables implemented in the utility and risk evaluation.

Table 6: Variables used for ROC evaluation in each dataset.

| Dataset | ROC Variables (Key/Categorical features) |
|---|---|
| UK | ECONPRIM, ETHGROUP, LTILL, QUALNUM, SEX, SOCLASS, TENURE, MSTATUS |
| Canada | ABIDENT, SEX, TENURE, URBAN, BPLMOM, BPLPOP, CITIZEN, LANG, MARST, RELATE, MINORITY, RELIG, BPL |
| Fiji | PROV, TENURE, RELATE, SEX, ETHNIC, MARST, RELIGION, BPLPROV, RESPROV, RESSTAT, SCHOOL, TRAVEL |
| Rwanda | STATUS, SEX, URBAN, OWNERSH, DISAB2, DISAB1, RELATE, RELIG, HINS, NATION, BPL |
| Indonesia | OWNERSHIP, LANDOWN, RELATE, SEX, MARST, HOMEMALE, RELIGION, SCHOOL, LIT, EDATTAIND, DISABLED |
| Adult | workclass, education, marital-status, occupation, relationship, race, sex, native-country, income |
| Churn | Geography, Gender, Tenure, NumOfProducts, HasCrCard, IsActiveMember, Exited |

## E  Additional and Detailed Results

### E.1 Results on Standard Tabular Generative Model Evaluations

Tables 10-13 report diagnostic metrics that complement the main Utility and Disclosure Risk scores. For WD, MMD, Shape, and Trend, *lower is better* as these metrics quantify discrepancies between real and synthetic statistics. For the C2ST detection score in Table 13, we follow the convention used in our evaluation code and report the detection score as defined there, where larger values correspond to stronger indistinguishability between real and synthetic samples.

Table 7: Variables used for CIO evaluation (targets and explanatory variables) in each dataset. Note that for CIO mainly uses logistic regression as the model, considering the categorical target variables, except the CreditScore and EstimatedSalary in Churn that uses linear regression since they are continuous variables.

| Dataset | Target Variables | Explanatory (Key) Variables |
|---|---|---|
| UK | TENURE, MSTATUS | ECONPRIM, ETHGROUP, LTILL, QUALNUM, SEX, SOCLASS, TENURE, MSTATUS, AGE |
| Canada | TENURE, MARST | ABIDENT, CLASSWK, DEGREE, EMPSTAT, SEX, URBAN, TENURE, MARST, AGE, HRSWK, INCTOT, WKSWORK |
| Fiji | TENURE, MARST | CLASSWKR, ETHNIC, RELIGION, EDATTAIN, SEX, PROV, TENURE, MARST, AGE |
| Rwanda | OWNERSH, MARST | DISAB1, EDCERT, CLASSWK, LIT, RELIG, SEX, OWNERSH, MARST, AGE |
| Indonesia | OWNERSHIP, MARST | LANDOWN, RELATE, SEX, HOMEFEM, HOMEMALE, RELIGION, LIT, SCHOOL, EDATTAIND, DISABLED, OWNERSHIP, MARST, AGE |
| Adult | income, marital-status | workclass, education-num, marital-status, occupation, relationship, race, sex, native-country, income, age, fnlwgt, capital-gain, capital-loss, hours-per-week |
| Churn | Exited, CreditScore, EstimatedSalary | Geography, Gender, Tenure, NumOfProducts, HasCrCard, IsActiveMember, Exited, CreditScore, Age, Balance, EstimatedSalary |

Table 8: Variables used for TCAP calculation (target and key classes) in each dataset.

| Dataset | Target Variables | Key Variables for Equivalence Class Definition |
|---|---|---|
| UK | TENURE, MSTATUS | ECONPRIM, ETHGROUP, LTILL, QUALNUM, SEX, SOCLASS, TENURE, MSTATUS |
| Canada | TENURE, MARST | ABIDENT, CLASSWK, DEGREE, EMPSTAT, SEX, URBAN, TENURE, MARST |
| Fiji | TENURE, MARST | CLASSWKR, ETHNIC, RELIGION, EDATTAIN, SEX, PROV, TENURE, MARST |
| Rwanda | OWNERSH, MARST | DISAB1, EDCERT, CLASSWK, LIT, RELIG, SEX, OWNERSH, MARST |
| Indonesia | OWNERSHIP, MARST | LANDOWN, RELATE, SEX, HOMEFEM, HOMEMALE, RELIGION, LIT, SCHOOL, EDATTAIND, DISABLED, OWNERSHIP, MARST |
| Adult | income, marital-status | workclass, education-num, marital-status, occupation, relationship, race, sex, native-country, income |
| Churn | Exited, CreditScore, EstimatedSalary | Geography, Gender, Tenure, NumOfProducts, HasCrCard, IsActiveMember, Exited |

**Discrepancy between data distributions (WD and MMD).** Tables 9 and 10 disseminated the results on MMD and WD. From Table 9, it seems that TabSyn dominates in two datasets (Fiji, Rwanda), while in Adult and Churn results show similar MMD values in four algorithms. Moreover, the difference of the distance of most algorithms, except TabDDPM are around 5 decimals. Therefore, MMD comparison results are inconclusive for this study, making WD becomes supporting results.

In contrast, Table 10 shows that **TabbyFlow-OT** achieves the lowest average per-continuous feature WD on five out of seven datasets (Canada, Fiji, Rwanda, Indonesia, and Adult), indicating the most faithful preservation of distributions in those settings. In contrast, **TabDDPM** is best on UK, and Churn. A con-

Table 9: Average MMD across 7 datasets. We report the mean $\pm$ standard deviation across 20 seeds of synthetic data generation. Bold and underline indicate the first and second best performing algorithms for each dataset, respectively. The table highlights the TabSyn are best in two datasets, while in Adult and Churn most algorithms are competitive, making it hard to conclude.

| Data | TabDDPM | TabSyn | TabSynFlow OT | TabSynFlow VP | TabbyFlow OT | TabbyFlow VP |
|------|---------|--------|---------------|---------------|--------------|--------------|
| UK | $\mathbf{4.0127 \times 10^{-4}}$ | $\underline{4.0197 \times 10^{-4}}$ | $4.0229 \times 10^{-4}$ | $4.0804 \times 10^{-4}$ | $4.0837 \times 10^{-4}$ | $4.1794 \times 10^{-4}$ |
| Canada | $2.1109 \times 10^{-2}$ | $5.7652 \times 10^{-4}$ | $\underline{5.6669 \times 10^{-4}}$ | $\mathbf{5.5739 \times 10^{-4}}$ | $5.9633 \times 10^{-4}$ | $5.9156 \times 10^{-4}$ |
| Fiji | $3.7338 \times 10^{-1}$ | $\mathbf{4.0162 \times 10^{-4}}$ | $4.0283 \times 10^{-4}$ | $4.0578 \times 10^{-4}$ | $\underline{4.0257 \times 10^{-4}}$ | $4.0270 \times 10^{-4}$ |
| Rwanda | $2.5037 \times 10^{-1}$ | $\mathbf{4.7202 \times 10^{-4}}$ | $4.9067 \times 10^{-4}$ | $5.5161 \times 10^{-4}$ | $\underline{4.9030 \times 10^{-4}}$ | $5.0168 \times 10^{-4}$ |
| Indonesia | $4.0213 \times 10^{-4}$ | $\underline{4.0174 \times 10^{-4}}$ | $\mathbf{4.0112 \times 10^{-4}}$ | $4.3640 \times 10^{-4}$ | $4.7961 \times 10^{-4}$ | $4.8380 \times 10^{-4}$ |
| Adult | $4.0026 \times 10^{-4}$ | $\mathbf{4.0005 \times 10^{-4}}$ | $\mathbf{4.0005 \times 10^{-4}}$ | $\mathbf{4.0005 \times 10^{-4}}$ | $\underline{4.0006 \times 10^{-4}}$ | $\mathbf{4.0005 \times 10^{-4}}$ |
| Churn | $4.0004 \times 10^{-4}$ | $\mathbf{4.0000 \times 10^{-4}}$ | $\mathbf{4.0000 \times 10^{-4}}$ | $\mathbf{4.0000 \times 10^{-4}}$ | $\underline{4.0001 \times 10^{-4}}$ | $\mathbf{4.0000 \times 10^{-4}}$ |

sistent pattern is that TabbyFlow variants have lower WD values than TabSynFlow-OT across all datasets, suggesting that overall TabbyFlow could preserve the data distributions better in these benchmarks.

Table 10: Average Wasserstein Distance across 7 datasets. We report the mean $\pm$ standard deviation across 20 seeds of synthetic data generation. Bold and underline indicate the first and second best performing algorithms for each dataset, respectively. The table highlights the domination of TabbyFlow as the best performers, followed by TabDDPM in three datasets, meaning that TabbyFlow is best in preserving continuous variables.

| Data | TabDDPM | TabSyn | TabSynFlow OT | TabSynFlow VP | TabbyFlow OT | TabbyFlow VP |
|------|---------|--------|---------------|---------------|--------------|--------------|
| UK | $\mathbf{1.7090\pm0.0024}$ | $1.7680\pm0.0026$ | $1.7690\pm0.0023$ | $1.8622\pm0.0028$ | $\underline{1.7177\pm0.0021}$ | $1.7675\pm0.0021$ |
| Canada | $8.9481\pm0.0179$ | $2.5620\pm0.0061$ | $2.5927\pm0.0047$ | $2.7237\pm0.0052$ | $\mathbf{2.3748\pm0.0053}$ | $\underline{2.4522\pm0.0072}$ |
| Fiji | $9.2802\pm0.0100$ | $2.5594\pm0.0035$ | $2.5869\pm0.0038$ | $2.7487\pm0.0039$ | $\mathbf{2.4620\pm0.0032}$ | $\underline{2.5274\pm0.0024}$ |
| Rwanda | $5.9976\pm0.0244$ | $0.7335\pm0.0029$ | $0.7483\pm0.0031$ | $0.7854\pm0.0022$ | $\mathbf{0.6797\pm0.0030}$ | $\underline{0.7154\pm0.0032}$ |
| Indonesia | $\underline{0.1732\pm0.0007}$ | $0.1757\pm0.0007$ | $0.1776\pm0.0006$ | $0.1859\pm0.0005$ | $\mathbf{0.1731\pm0.0006}$ | $0.1778\pm0.0006$ |
| Adult | $0.8392\pm0.0024$ | $0.7976\pm0.0024$ | $0.8141\pm0.0023$ | $0.8558\pm0.0025$ | $\mathbf{0.7665\pm0.0023}$ | $\underline{0.7937\pm0.0026}$ |
| Churn | $\mathbf{0.1528\pm0.0027}$ | $\underline{0.1592\pm0.0021}$ | $0.1600\pm0.0020$ | $0.1930\pm0.0031$ | $0.1653\pm0.0023$ | $0.1652\pm0.0022$ |

**Mixed-type univariate fidelity (Shape).** The Shape metric aggregates column-wise discrepancies across both continuous and categorical attributes. Table 11 indicates a split regime: TabbyFlow is strongest on UK, Canada, Fiji, and Rwanda, which aligns with its strong categorical TVD performance and its ability to preserve mixed-type univariate structure. TabSynFlow-OT is strongest on Indonesia, Adult, and Churn, which aligns with its consistently low WD and indicates that these datasets benefit more from improvements in continuous modeling or from the specific trajectory specification used by TabSynFlow-OT.

**Mixed-type pairwise fidelity (Trend).** Trend evaluates how well pairwise dependence is preserved, combining correlation discrepancies for continuous pairs and contingency-table discrepancies for categorical pairs. Table 12 largely mirrors the Shape findings. TabbyFlow achieves the lowest Trend error on UK, Canada, Fiji, and Rwanda, indicating improved preservation of pairwise structure on these datasets. However, TabSynFlow-OT is best on Indonesia, Adult, and Churn, and TabbyFlow performs poorly on Churn in particular, where its Trend error is several times larger than competing methods. Overall, these results suggest that TabbyFlow's dependence preservation is dataset-dependent, and its strongest gains occur on the census-style datasets with richer categorical structure, while TabSynFlow-OT provides more stable pairwise behavior on Adult, Churn, and Indonesia.

**Detectability (C2ST).** Table 13 reports C2ST detection scores using logistic regression. Across most datasets, TabbyFlow variants achieve the strongest detection scores, consistent with producing samples that are harder to distinguish from real data under the chosen detector. There are notable exceptions. On

Table 11: Performance comparison on the error rates (%) of Shape. We report the mean ± standard deviation across 20 seeds of synthetic data generation. Bold and underline indicate the first and second best performing algorithms for each dataset, respectively. The table highlights the domination of TabbyFlow and TabSynFlow as the best performers, meaning that the algorithms are best in preserving the distribution of mixed-type data.

| Data | TabDDPM | TabSyn | TabSynFlow OT | TabSynFlow VP | TabbyFlow OT | TabbyFlow VP |
|---|---|---|---|---|---|---|
| UK | 0.95±0.04 | 0.93±0.04 | 0.71±0.03 | 1.41±0.04 | **0.58±0.04** | 0.62±0.03 |
| Canada | 40.58±0.12 | 3.30±0.07 | 3.74±0.04 | 4.45±0.05 | 3.22±0.04 | **3.17±0.05** |
| Fiji | 50.74±0.12 | 1.04±0.04 | 1.03±0.03 | 2.21±0.04 | 0.80±0.05 | **0.75±0.06** |
| Rwanda | 24.28±0.12 | 0.94±0.05 | 0.82±0.04 | 1.45±0.05 | 0.56±0.04 | **0.51±0.05** |
| Indonesia | 0.34±0.03 | 0.46±0.03 | **0.21±0.03** | 0.40±0.03 | 0.29±0.04 | 0.37±0.03 |
| Adult | 1.56±0.03 | 0.99±0.04 | **0.80±0.04** | 1.46±0.06 | 1.22±0.05 | 1.11±0.05 |
| Churn | 1.16±0.08 | 1.04±0.13 | **0.98±0.07** | 2.31±0.07 | 2.72±0.09 | 2.55±0.10 |

Table 12: Performance comparison on the error rates (%) of Trend. We report the mean ± standard deviation across 20 seeds of synthetic data generation. Bold and underline indicate the first and second best performing algorithms for each dataset, respectively. The table highlights the domination of TabbyFlow and TabSynFlow as the best performers, meaning that the algorithms are best in preserving the correlation and contingency of mixed-type data.

| Data | TabDDPM | TabSyn | TabSynFlow OT | TabSynFlow VP | TabbyFlow OT | TabbyFlow VP |
|---|---|---|---|---|---|---|
| UK | 1.79±0.05 | 1.91±0.04 | 1.66±0.03 | 3.17±0.05 | **1.38±0.05** | 1.81±0.06 |
| Canada | 61.3±0.13 | 8.05±0.52 | 8.71±0.30 | 8.15±0.23 | **7.12±0.32** | 7.29±0.30 |
| Fiji | 78.47±0.05 | 1.96±0.04 | 2.10±0.04 | 3.74±0.05 | **1.56±0.06** | 1.58±0.06 |
| Rwanda | 45.19±0.21 | 1.37±0.09 | 1.43±0.10 | 2.18±0.06 | **1.01±0.08** | 1.14±0.13 |
| Indonesia | 0.61±0.05 | 0.78±0.04 | **0.43±0.04** | 0.77±0.04 | 0.72±0.08 | 0.95±0.07 |
| Adult | 3.64±0.06 | 2.11±0.07 | **2.07±0.25** | 3.02±0.17 | 2.82±0.45 | 2.75±0.44 |
| Churn | 2.38±0.15 | 2.41±0.25 | **2.22±0.15** | 4.60±0.15 | 8.00±0.11 | 6.96±0.11 |

Indonesia, TabSynFlow-OT attains the top score, and on Churn, TabSyn and TabSynFlow-OT outperform TabbyFlow. TabDDPM is highly inconsistent across datasets, with extremely low scores on Canada and Fiji and much higher scores elsewhere, indicating substantial dataset sensitivity for this detector-based criterion.

Table 13: Performance comparison on the detection score (C2ST) using logistic regression classifier. We report the mean ± standard deviation across 20 seeds of synthetic data generation. Bold and underline indicate the first and second best performing algorithms for each dataset, respectively. The table highlights the domination of TabbyFlow as the best performers, meaning that the algorithms are best in producing realistic synthetic data.

| Data | TabDDPM | TabSyn | TabSynFlow OT | TabSynFlow VP | TabbyFlow OT | TabbyFlow VP |
|---|---|---|---|---|---|---|
| UK | 0.9563 | 0.9561 | 0.9652 | 0.8801 | 0.9789 | **0.9827** |
| Canada | 0.0555 | 0.8439 | 0.8404 | 0.8530 | **0.8770** | 0.8712 |
| Fiji | 0.0376 | 0.9763 | 0.9815 | 0.9550 | 0.9821 | **0.9859** |
| Rwanda | 0.4347 | 0.9824 | 0.9779 | 0.9632 | 0.9932 | **0.9972** |
| Indonesia | 0.9920 | 0.9891 | **0.9983** | 0.9887 | 0.9898 | 0.9808 |
| Adult | 0.9427 | 0.9704 | 0.9826 | 0.9458 | 0.9639 | **0.9881** |
| Churn | 0.9748 | **0.9957** | 0.9949 | 0.8700 | 0.9222 | 0.9270 |

**Sample quality and coverage ($\alpha$-precision and $\beta$-recall).** Tables 14–15 report $\alpha$-precision (sample fidelity) and $\beta$-recall (coverage) averaged over 20 generation seeds. Across all datasets, $\alpha$-precision is high for TabSynFlow and TabbyFlow, indicating that their synthetic records typically fall within high-density regions of the real data. TabSynFlow is best on four datasets (Fiji, Rwanda, Adult, and Churn) and second-best on two (UK and Indonesia), while TabbyFlow is best on three datasets (UK, Canada, and Indonesia) and frequently appears among the top two, showing that both families produce high-fidelity samples.

In contrast, $\beta$-recall reveals clearer dataset dependence. TabbyFlow achieves the best coverage on five datasets (UK, Canada, Fiji, Rwanda, and Adult), suggesting that it more consistently captures diverse modes of the real distribution rather than concentrating on a narrow high-density region. However, Indonesia and Churn are exceptions: on Indonesia, TabSyn and TabSynFlow are slightly higher than TabbyFlow, and on Churn, TabDDPM achieves the highest $\beta$-recall. Overall, these results indicate that TabSynFlow and TabbyFlow are comparably strong in sample-level fidelity, while TabbyFlow more often improves coverage, with a small number of datasets where alternative methods provide better recall.

Table 14: Performance comparison on $\alpha$-precision (higher is better). We report mean $\pm$ standard deviation across 20 synthetic generation seeds. Bold and underline indicate the best and second-best methods per dataset. Overall, TabSynFlow and TabbyFlow consistently achieve the highest $\alpha$-precision, indicating strong sample-level fidelity across datasets.

| Data | TabDDPM | TabSyn | TabSynFlow OT | TabSynFlow VP | TabbyFlow OT | TabbyFlow VP |
|---|---|---|---|---|---|---|
| UK | 96.95±0.15 | 98.67±0.21 | 99.05±0.16 | **99.50±0.11** | 99.10±0.16 | 98.82±0.21 |
| Canada | 30.20±0.17 | 98.14±0.30 | 98.65±0.22 | 98.18±0.16 | 98.67±0.21 | **98.90±0.24** |
| Fiji | 50.99±0.03 | 98.46±0.15 | 98.72±0.18 | **99.47±0.09** | 98.64±0.19 | 99.25±0.17 |
| Rwanda | 49.93±0.37 | 97.97±0.32 | **99.11±0.15** | 98.24±0.15 | 98.85±0.26 | 99.03±0.30 |
| Indonesia | 98.96±0.12 | 98.50±0.11 | 99.49±0.11 | 99.35±0.11 | 99.41±0.12 | **99.66±0.07** |
| Adult | 95.42±0.17 | 98.50±0.22 | **99.33±0.12** | 99.03±0.10 | 99.27±0.24 | 99.19±0.26 |
| Churn | 96.87±0.37 | 98.56±0.37 | **98.99±0.44** | 97.09±0.52 | 98.52±0.39 | 98.81±0.39 |

Table 15: Performance comparison on $\beta$-recall (higher is better). We report mean $\pm$ standard deviation across 20 synthetic generation seeds. Bold and underline indicate the best and second-best methods per dataset. TabbyFlow attains the highest $\beta$-recall on most datasets, indicating stronger coverage of the real data distribution

| Data | TabDDPM | TabSyn | TabSynFlow OT | TabSynFlow VP | TabbyFlow OT | TabbyFlow VP |
|---|---|---|---|---|---|---|
| UK | 68.84±0.13 | 67.58±0.15 | 67.22±0.12 | 62.92±0.15 | **69.47±0.18** | 67.30±0.25 |
| Canada | 0.45±0.04 | 33.39±0.19 | 32.47±0.28 | 28.66±0.24 | **38.62±0.29** | 35.83±0.30 |
| Fiji | 0.02±0.01 | 56.79±0.17 | 55.18±0.19 | 49.28±0.25 | **59.78±0.17** | 57.79±0.24 |
| Rwanda | 0.87±0.28 | 84.52±0.20 | 84.18±0.22 | 82.90±0.20 | **86.26±0.18** | 85.24±0.19 |
| Indonesia | 96.35±0.06 | **96.47±0.04** | 96.46±0.07 | 96.36±0.07 | 95.27±0.15 | 95.16±0.14 |
| Adult | 45.35±0.19 | 48.78±0.28 | 47.57±0.18 | 45.58±0.24 | **50.22±0.25** | 48.93±0.22 |
| Churn | **54.22±0.67** | 50.93±0.51 | 50.96±0.45 | 51.33±0.34 | 48.30±0.43 | 49.01±0.67 |

## E.2 Number of Function Evaluations

Figure 3 shows how computational cost is related to the quality of the output between methods (TabSyn, TabSynFlow and TabbyFlow) and datasets. In general, both utility and risk increase as NFEs grow. In four datasets (UK, RW, CA, FI, ID), TabbyFlow seems to outperform TabSyn. However, CH, TabSynFlow-OT outperforms TabSyn. Therefore, if one has a limited computational budget, they may choose TabbyFlow or TabSynFlow using the OT path as the algorithm that performed best on low-step setting among all algorithms, particularly TabSyn.

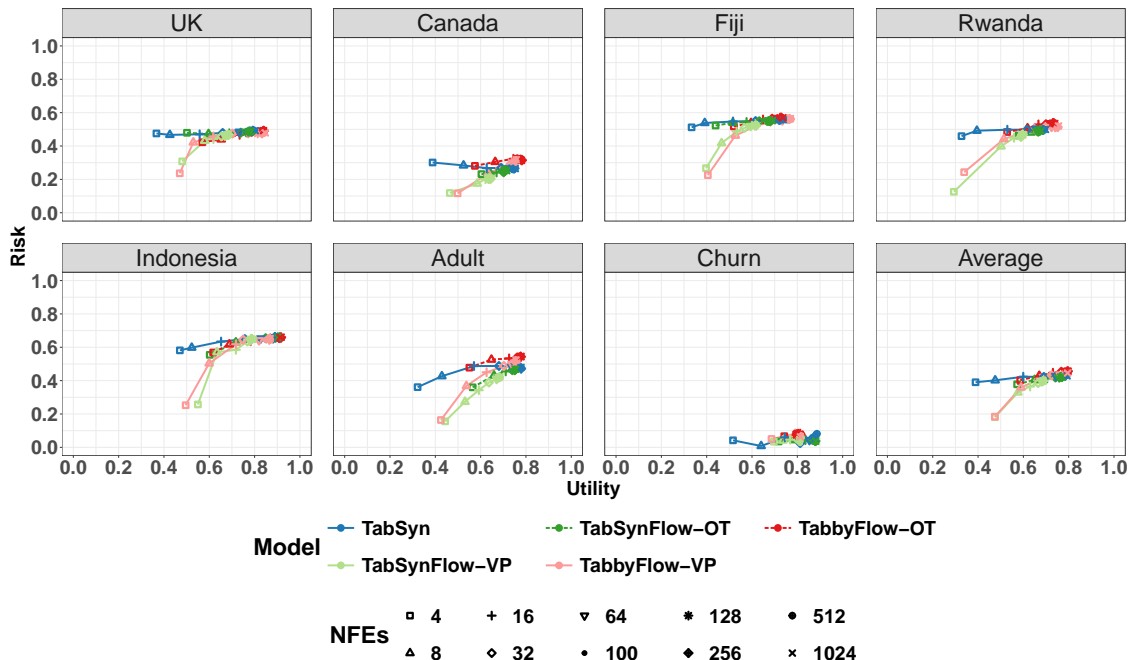

Figure 3: Utility and risk evaluation of TabSynFlow and TabbyFlow based on dataset and average using different number of evaluations with formula $2^n, n = [2, ..., 10]$. TabbyFlow-OT (and TabSynFlow-OT) achieve strong utility at low NFEs ($\leq 100$) and tend to converge by $\approx 128$, whereas TabSyn requires higher NFEs to catch up—so OT + flow-matching is attractive under tight compute budgets.

### E.3 Integration Time

While the original implementation of flow matching uses full ODE integration, it is possible to terminate the ODE at intermediate times—a principle explored in this section. Our analysis of ODE integration times ($t_{ode}$) in Figure 4, spanning datasets including Rwanda, Adult, Churn, UK, Indonesia, Canada, and Fiji, offers critical insights into how utility and risk evolve during synthetic data generation. For TabSynFlow with optimal transport (OT), early stop at $t_{ode} = 0.6$ consistently produces high-utility synthetic data, whereas variance preserving (VP) paths in both TabSynFlow and TabbyFlow tend to underperform at this stage. This indicates that OT path better retain the data structure in earlier integration steps, while VP introduces noise prematurely. Notably, the Rwanda dataset presents an interesting anomaly: TabbyFlow-OT achieves slightly better utility than VP methods at $t_{ode} = 0.6$, which may be attributed to the dataset's relatively simple feature distributions.

Figure 5 showed the utility and risk in details when $0.9 \leq t_{ode} \leq 1$. At $t = 0.9$, OT methods maintain superior performance, particularly for the UK and Adult datasets, although the gap between OT and VP narrows for the latter. This implies dataset-specific sensitivity to VP's noise characteristics. Near full integration ($t_{ode} \rightarrow 1.0$), all algorithms exhibit erratic behaviour in some datasets such as Fiji: the utility drops and the risk increases. This problem suggests that flow matching is prone to quality drops when fully integrated.

## F   Flow Matching Results on Different Interpolants

### F.1   Interpolation Path in Flow Matching

A central element in continuous-time generative modelling is the choice of *interpolation schedule*, which determines how the clean data $x_1$ is progressively transformed into noise, and conversely how the noise is mapped back to the data. This schedule is characterised by two functions: the signal decay factor $\alpha(t)$ and

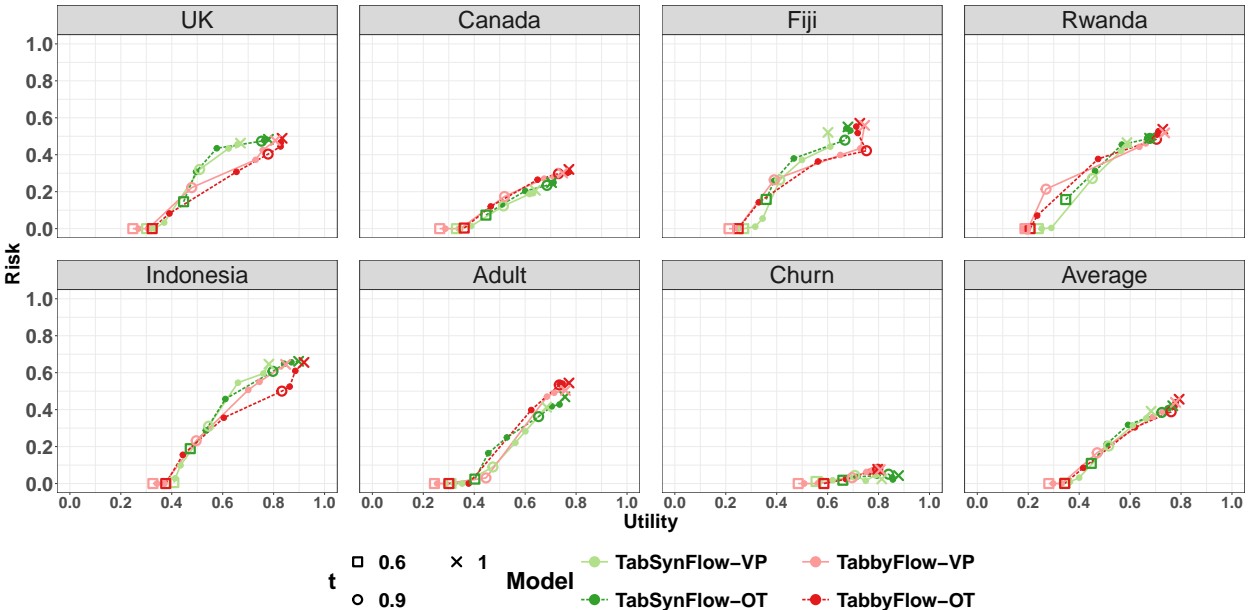

Figure 4: Effect of ODE integration time on utility–risk (per dataset + average) for TabSynFlow and TabbyFlow. Utility and risk evaluation of TabSynFlow and TabbyFlow based on dataset and average on different integration time from $t_{\text{ode}} = [0.6, 1]$ with interval 0.1. Colors encode method and path (VP vs OT). OT achieves high-utility solutions early (e.g., $t_{\text{ode}} = 0.6$), while pushing to $t_{\text{ode}} \to 1$ often reduces utility and/or increases risk—so early stopping is preferable.

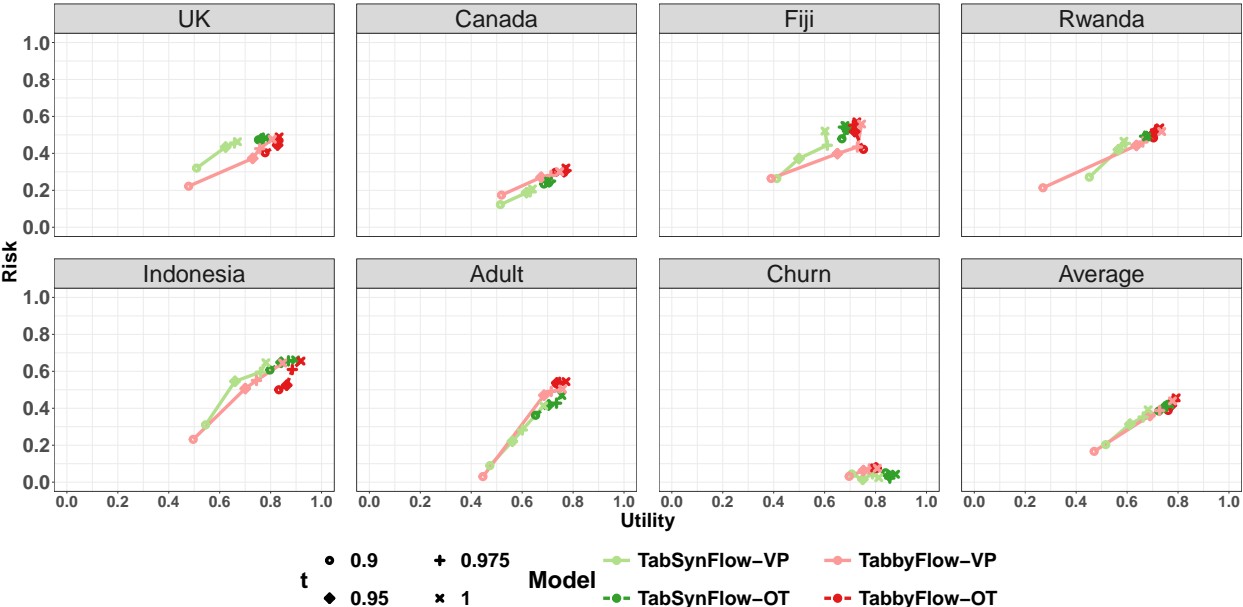

Figure 5: Utility and risk evaluation of TabSynFlow and TabbyFlow based on dataset and average on late integration time $t = [0.9, 0.95, 0.975, 1]$. For selected datasets, we zoom into the late integration window using the same color/marker scheme as Figure 4. Take-home: Near full integration, instability emerges across several datasets (e.g., Fiji), consistent with utility drops and risk increases; OT remains comparatively robust but still degrades as $t_{\text{ode}} \to 1$ in some cases.

the noise scale $\sigma(t)$. Together, these define the conditional distribution.

$$x_t|x_1 \sim \mathcal{N}\big(\alpha(t)x_1, \sigma^2(t)I\big),$$

as well as their time derivatives $\dot{\alpha}(t)$ and $\dot{\sigma}(t)$, which govern the dynamics in both ODE and SDE formulations.

## F.2 Interpolants in Flow Matching

There are two commonly used interpolants to define the probability path in flow matching.

The first is the **optimal transport (OT)** path (Liu et al. 2022; Lipman et al. 2023), where the mean and standard deviation of the conditional distribution evolve linearly over time, resulting in straight-line (conditional) paths connecting the noise and data distributions. OT-based trajectories are computationally straightforward, efficient, and easy to parameterise due to their time-invariant structure, as shown in equation F.1

$$x_t = tx_1 + (1 - (1 - \sigma_{\min})t)x_0$$
$$u_t(x_t \mid x_1) = \frac{x_1 - (1 - \sigma_{\min})x_t}{1 - (1 - \sigma_{\min})t}, \tag{F.1}$$

where $\sigma_{\min}$ is added to ensure the target density has non-zero measure everywhere (or: we can imagine first convolving the data distribution with a small Gaussian).

The second type of interpolant is the **variance-preserving (VP)** path, derived from stochastic differential equations commonly used in diffusion models (Lipman et al. 2023). In this approach, noise is gradually added to the data during the forward process, while the reverse process learns to de-noise using time-dependent vector fields. In contrast to the straight OT path, where the conditional mean evolves linearly in $t$, the VP path follows a noise schedule that induces non-uniform change over time, with smaller updates in early times and larger updates later. The parameters $\alpha_t$ and $\sigma_t$ are determined by cumulative functions of the noise scale function $\beta(t) = \beta_{\min} + t(\beta_{\max} - \beta_{\min})$, as given in equation F.2.

$$\alpha_t = e^{-\frac{1}{2}T(t)}, \quad T(t) = \int_0^{1-t} \beta(s)ds, \quad \sigma_t = \sqrt{1 - \alpha_t^2}$$
$$u_t(x_t|x_1) = -\frac{\dot{T}(t)}{2} \left[ \frac{e^{-T(t)}x_t - e^{-\frac{1}{2}T(t)}x_1}{1 - e^{-T(t)}} \right] \tag{F.2}$$

The choice of trajectory – or interpolant – significantly affects data synthesis performance, which we explore in this paper. Depending on the data structure or modelling objectives, either trajectory can be more suitable. Therefore, flow matching formulations can also use alternative paths such as variance-exploding (Lipman et al. 2023), cosine (Albergo & Vanden-Eijnden 2023), or logit-normal (Black Forest Labs 2025) interpolants. In this section, we summarise these schedules in two comparative tables: Table 16 for the interpolants $(\alpha(t), \sigma(t))$ and Table 17 for their derivatives $(\dot{\alpha}(t), \dot{\sigma}(t))$.

## F.3 Empirical Results on TabSynFlow

Table 18 disseminated the simulation results of TabSynFlow using different paths. Based on the table, it seems that the comparative analysis shows clear distinctions between the paths. optimal transport (OT) offers the best balance of utility and risk. variance preserving (VP) and Cosine paths reduce risk further, making them well suited for privacy-sensitive contexts, though at a slight cost to utility. In contrast, the variance exploding (VE) and Logit paths underperforms on both measures. Overall, OT is a strong default, VP is preferable when privacy is prioritised, and Cosine remains a viable alternative, although path choice should be validated for each dataset.

## F.4 Empirical Results on TabbyFlow

The results on Table 19 highlights clear differences of the TabbyFlow results between paths. optimal transport (OT) delivers strong overall performance and serves as a reliable default, while Logit performs comparably

Table 16: Interpolants $\alpha(t)$ and $\sigma(t)$ for different schedules.

| Interpolation | $\alpha(t)$ | $\sigma(t)$ |
|---|---|---|
| Linear | $1 - t$ | $t$ |
| Variance-Preserving | $\exp\left(-\frac{1}{2}\int_0^{1-t}\beta(s)\,ds\right)$ | $\sqrt{1-\alpha^2(t)}$ |
| Variance-Exploding | $1$ | $\sigma_{\min}\left(\frac{\sigma_{\max}}{\sigma_{\min}}\right)^t$ |
| Cosine | $\sin\left(\frac{\pi}{2}t\right)$ | $\cos\left(\frac{\pi}{2}t\right)$ |
| Logit-Normal | $\dfrac{e^\mu}{e^\mu + (\frac{1}{ct}-1)^\lambda}$ $\mu = \log 3,\ \lambda = 1$ | $1 - \alpha(t)$ |

Table 17: Derivative of interpolants for different schedules.

| Interpolation | $\dot\alpha(t)$ | $\dot\sigma(t)$ |
|---|---|---|
| Linear | $-1$ | $1$ |
| Variance-Preserving | $-\frac{1}{2}\beta(t)\alpha(t)$ | $\frac{1}{2}\dfrac{\beta(t)\alpha^2(t)}{\sigma(t)}$ |
| Variance-Exploding | $0$ | $\sigma(t)\ln\left(\frac{\sigma_{\max}}{\sigma_{\min}}\right)$ |
| Cosine | $\frac{\pi}{2}\cos\left(\frac{\pi}{2}t\right)$ | $-\frac{\pi}{2}\sin\left(\frac{\pi}{2}t\right)$ |
| Logit-Normal | $\dfrac{\lambda\exp(\mu)}{ct^2(e^\mu + (\frac{1}{ct}-1)^\lambda)^2}\left(\frac{1}{ct}-1\right)^{\lambda-1}$ | $-\dot\alpha(t)$ |

Table 18: Comparison of utility and risk across different paths on TabSynFlow.

| Dataset | OT | | VP | | VE | | Cosine | | Logit-Normal | |
|---|---|---|---|---|---|---|---|---|---|---|
| | Utility | Risk | Utility | Risk | Utility | Risk | Utility | Risk | Utility | Risk |
| UK | 0.7796 | 0.4834 | 0.6696 | 0.4630 | 0.2387 | 0.5263 | 0.7554 | 0.4802 | 0.5435 | 0.4622 |
| Canada | 0.7035 | 0.2493 | 0.6403 | 0.2073 | 0.2841 | 0.3827 | 0.7021 | 0.2485 | 0.4875 | 0.2233 |
| Fiji | 0.6798 | 0.5506 | 0.6014 | 0.5208 | 0.2800 | 0.5376 | 0.7115 | 0.5526 | 0.3465 | 0.4593 |
| Rwanda | 0.6765 | 0.4894 | 0.5875 | 0.4655 | 0.2723 | 0.5250 | 0.6681 | 0.4923 | 0.4528 | 0.4513 |
| Indonesia | 0.8981 | 0.6616 | 0.7814 | 0.6464 | 0.4467 | 0.6186 | 0.8732 | 0.6547 | 0.5206 | 0.4782 |
| Adult | 0.7560 | 0.4693 | 0.6843 | 0.4131 | 0.2745 | 0.5023 | 0.7466 | 0.4675 | 0.4429 | 0.2489 |
| Churn | 0.8784 | 0.0429 | 0.8126 | 0.0246 | 0.4922 | 0.0216 | 0.8284 | 0.0562 | 0.7189 | 0.0630 |
| Average | 0.7674 | 0.4209 | 0.6824 | 0.3915 | 0.3269 | 0.4449 | 0.7551 | 0.4217 | 0.5018 | 0.3409 |

and in some cases slightly better. VP and Cosine stand out as the least risk, making them suitable when privacy is prioritised, though they sacrifice a little utility. Variance exploding (VE), by contrast, shows worst performance, limiting its general usefulness. These findings highlight that trajectory choice should align with application needs: OT offers balanced performance, VP prioritises privacy protection, while Logit and cosine present viable alternatives to OT and VP, respectively.

## F.5 SDE Performance on Different $g_t$

Tables 20 and 21 disseminate the comparison of the difference in utility ($\Delta$U) and risk ($\Delta$R) when using SDE with different $g_t$ on TabbyFlow compared to its ODE counterparts as a baseline (See Table 19). The tables

Table 19: Comparison of Utility and Risk Across Different Paths on TabbyFlow. We report the mean across 20 seeds of synthetic data generation.

| Dataset | OT | | VP | | VE | | Cosine | | Logit-Normal | |
|---|---|---|---|---|---|---|---|---|---|---|
| | Utility | Risk | Utility | Risk | Utility | Risk | Utility | Risk | Utility | Risk |
| UK | 0.8333 | 0.4889 | 0.8066 | 0.4765 | 0.3238 | 0.5331 | 0.8001 | 0.4721 | 0.8201 | 0.4887 |
| Canada | 0.7718 | 0.3206 | 0.7472 | 0.3008 | 0.4083 | 0.3437 | 0.7626 | 0.3102 | 0.7619 | 0.3032 |
| Fiji | 0.7263 | 0.5705 | 0.7451 | 0.5588 | 0.3052 | 0.6000 | 0.7121 | 0.5341 | 0.7303 | 0.5694 |
| Rwanda | 0.7284 | 0.5379 | 0.7358 | 0.5180 | 0.3267 | 0.5222 | 0.7440 | 0.5138 | 0.7396 | 0.5271 |
| Indonesia | 0.9191 | 0.6556 | 0.8473 | 0.6447 | 0.5309 | 0.5922 | 0.7775 | 0.6338 | 0.8984 | 0.6365 |
| Adult | 0.7720 | 0.5443 | 0.7581 | 0.5037 | 0.3097 | 0.5666 | 0.7205 | 0.5107 | 0.7665 | 0.5580 |
| Churn | 0.7966 | 0.0773 | 0.8066 | 0.0753 | 0.5563 | 0.0481 | 0.7913 | 0.0841 | 0.7982 | 0.0767 |
| Average | 0.7565 | 0.4215 | 0.7498 | 0.4036 | 0.3831 | 0.4299 | 0.7375 | 0.4015 | 0.7526 | 0.4160 |

reveal consistently small deviations across all datasets, with the majority of differences within $[-0.02, +0.02]$. This minimal variation strongly validates the theoretical equivalence between the SDE and ODE marginals, demonstrating that SDEs preserve the same distributional properties regardless of the specific $g_t$ function employed. The results confirm that the controlled stochasticity introduced through SDE sampling maintains distributional fidelity while offering additional flexibility in the sampling process.

In particular, several configurations show the desirable outcome of simultaneously improving both utility and reducing risk. For example, Rwanda in the OT configuration achieves $\Delta U = +0.0105$ and $\Delta R = -0.0054$, while Fiji in the VP configuration achieves $\Delta U = +0.0067$ and $\Delta R = -0.0017$. These dual improvements indicate that SDE-based approaches can not only match the performance of ODE but, in some cases, produced high utility data with lower privacy risk. This dataset-dependent behavior underscores the importance of empirical selection for $g_t$ in synthesiser design, as the configuration can vary based on feature complexity, data distribution, or presence of rare categories.

Table 20: Difference of utility and risk across different $g_t$ setting in SDE w.r.t. the ODE results on TabbyFlow-OT. We report the mean across 20 seeds of synthetic data generation.

| Dataset | OT | | VP | | VE | | Cos | | Logit-Normal | |
|---|---|---|---|---|---|---|---|---|---|---|
| | $\Delta U(\uparrow)$ | $\Delta R(\downarrow)$ | $\Delta U(\uparrow)$ | $\Delta R(\downarrow)$ | $\Delta U(\uparrow)$ | $\Delta R(\downarrow)$ | $\Delta U(\uparrow)$ | $\Delta R(\downarrow)$ | $\Delta U(\uparrow)$ | $\Delta R(\downarrow)$ |
| UK | -0.0073 | -0.0011 | -0.0044 | 0.0000 | -0.0066 | -0.0007 | -0.0024 | 0.0003 | -0.0103 | 0.0022 |
| Canada | -0.0011 | -0.0020 | 0.0060 | 0.0007 | 0.0004 | -0.0026 | 0.0047 | -0.0010 | 0.0034 | -0.0076 |
| Fiji | 0.0006 | -0.0024 | 0.0067 | -0.0017 | 0.0042 | -0.0027 | 0.0010 | -0.0052 | 0.0029 | -0.0001 |
| Rwanda | 0.0105 | -0.0054 | 0.0142 | 0.0004 | 0.0093 | -0.0050 | 0.0123 | -0.0039 | 0.0112 | -0.0023 |
| Indonesia | 0.0052 | -0.0082 | -0.0142 | -0.0004 | -0.0020 | 0.0011 | 0.0041 | 0.0016 | 0.0007 | 0.0012 |
| Adult | 0.0077 | 0.0012 | 0.0027 | 0.0040 | 0.0047 | 0.0021 | 0.0067 | 0.0038 | 0.0039 | 0.0045 |
| Churn | 0.0036 | -0.0156 | 0.0021 | 0.0070 | 0.0025 | -0.0047 | 0.0024 | 0.0112 | 0.0032 | 0.0210 |

# G   Ablation Studies

## G.1   Variance Comparison Results

The comparative analysis of utility and risk across datasets in Table 22 highlights the performance distinction between the theory-based and our approaches. In most cases, our method consistently achieves higher utility scores than the theory method, indicating better preservation of data usefulness for downstream tasks. For instance, in the UK dataset, our approach yields a utility of $0.8333 \pm 0.0191$ compared to $0.5137 \pm 0.0072$ for the theory method. However, higher utility in our approach is often accompanied by slight increased risk values. For example, the risk for our approach in the UK dataset is $0.4889 \pm 0.0043$, higher than that of the

Table 21: Difference of utility and risk across different $g_t$ setting in SDE w.r.t. the ODE results on TabbyFlow-VP. We report the mean across 20 seeds of synthetic data generation.

| Dataset | OT | | VP | | VE | | Cos | | Logit-Normal | |
|---|---|---|---|---|---|---|---|---|---|---|
| | $\Delta U(\uparrow)$ | $\Delta R(\downarrow)$ | $\Delta U(\uparrow)$ | $\Delta R(\downarrow)$ | $\Delta U(\uparrow)$ | $\Delta R(\downarrow)$ | $\Delta U(\uparrow)$ | $\Delta R(\downarrow)$ | $\Delta U(\uparrow)$ | $\Delta R(\downarrow)$ |
| UK | -0.0035 | -0.0005 | -0.0070 | -0.0024 | -0.0101 | -0.0020 | -0.0072 | -0.0022 | -0.0045 | -0.0010 |
| Canada | 0.0015 | 0.0029 | 0.0025 | -0.0027 | 0.0014 | 0.0017 | 0.0026 | 0.0007 | 0.0048 | 0.0076 |
| Fiji | 0.0038 | -0.0012 | 0.0011 | -0.0017 | 0.0097 | -0.0022 | 0.0021 | -0.0026 | 0.0095 | -0.0006 |
| Rwanda | 0.0103 | 0.0012 | 0.0038 | 0.0000 | 0.0032 | -0.0030 | 0.0084 | 0.0054 | 0.0098 | -0.0017 |
| Indonesia | 0.0135 | 0.0009 | 0.0137 | 0.0007 | 0.0078 | 0.0009 | 0.0188 | 0.0027 | 0.0098 | -0.0030 |
| Adult | 0.0043 | -0.0035 | 0.0082 | -0.0021 | -0.0006 | -0.0028 | 0.0055 | 0.0010 | 0.0004 | 0.0006 |
| Churn | 0.0001 | -0.0102 | 0.0002 | 0.0018 | 0.0009 | -0.0014 | -0.0009 | 0.0061 | -0.0011 | -0.0022 |

theoretical variance ($0.3998 \pm 0.0074$), but still having significantly higher utility. Meanwhile, the only better performance of the theoretical approach is in Rwanda (theoretical: U=$0.7374 \pm 0.0155$, R=$0.5019 \pm 0.0125$; ours: U=$0.7284 \pm 0.0231$, R=$0.5379 \pm 0.0139$). Overall, our proposed approach generally provides superior performance across most datasets.

Table 22: Comparative performance of synthetic data results using theoretical (left) and relaxed (right) variance for TabbyFlow-OT. We report the mean $\pm$ standard deviation across 20 seeds of synthetic data generation. The table highlights that relaxing the theoretical variance to learn the distribution of continuous variables frequently leads to better performance.

| dataset | $0.5A_t(x_t)^{-2}$ | | $0.5A_t(x_t)^{-1}$ (**ours**) | |
|---|---|---|---|---|
| | Utility | Risk | Utility | Risk |
| UK | $0.5137 \pm 0.0072$ | $0.3998 \pm 0.0074$ | $0.8333 \pm 0.0191$ | $0.4889 \pm 0.0043$ |
| Canada | $0.7479 \pm 0.0109$ | $0.2961 \pm 0.0175$ | $0.7718 \pm 0.0211$ | $0.3206 \pm 0.0146$ |
| Fiji | $0.7002 \pm 0.0228$ | $0.5574 \pm 0.0060$ | $0.7263 \pm 0.0205$ | $0.5705 \pm 0.0063$ |
| Rwanda | $0.7319 \pm 0.0199$ | $0.5019 \pm 0.0125$ | $0.7284 \pm 0.0231$ | $0.5379 \pm 0.0139$ |
| Indonesia | $0.6076 \pm 0.0076$ | $0.6139 \pm 0.0145$ | $0.9191 \pm 0.0139$ | $0.6556 \pm 0.0133$ |
| Adult | $0.7374 \pm 0.0155$ | $0.4917 \pm 0.0162$ | $0.7720 \pm 0.0231$ | $0.5443 \pm 0.0177$ |
| Churn | $0.7613 \pm 0.0103$ | $0.0635 \pm 0.0792$ | $0.7966 \pm 0.0098$ | $0.0773 \pm 0.0579$ |

## G.2 Stress-test: Latent-space vs. Data-space Flow Matching.

### G.2.1 High Cardinality Data: Diabetes

Table 23 reports an ablation on the Diabetes dataset ($D_{\mathrm{num}} = 8$, $\max(K_d) = 790$, $\sum K_d = 2513$) [3] comparing latent-space flow matching (TabSynFlow) and data-space flow matching (TabbyFlow). Due to the memory constraint in high dimensional setting, we excluded the high cardinality columns when calculating $\alpha$-precision, $\beta$-recall, and WD, and also do batch-split on $\alpha$-precision, $\beta$-recall.

Based on the table, TabbyFlow is stronger at preserving data utility and fidelity, with lower WD, shape, and trend values and higher $\alpha$-precision and $\beta$-recall. In contrast, TabSynFlow better matches continuous marginals, achieving substantially lower WD, and TabSynFlow-OT and TabbyFlow-VP attains the most favorable utility-risk balance among the four configurations. Overall, this stress test suggests a clear division of strengths: with the availability of high category data, data-space VFM (TabbyFlow) improves utility and fidelity, while latent-space FM (TabSynFlow) improves continuous alignment.

---

[3]Data can be accessed in `https://archive.ics.uci.edu/dataset/296/diabetes+130-us+hospitals+for+years+1999-2008`

Table 23: High-cardinality stress test on the Diabetes dataset ($n = 101,766$, $D_{\text{num}} = 8$, $\max(K_d) = 790$, $\sum K_d = 2513$): comparison of latent-space flow matching (TabSynFlow) and data-space flow matching (TabbyFlow) under OT and VP trajectories. We report the mean $\pm$ standard deviation across 20 seeds of synthetic data generation.. The results show that TabbyFlow performed better in both OT and VP, while in latent space, TabSynFlow-OT showed best performance.

| Evaluation | TabSynFlow-OT | TabSynFlow-VP | TabbyFlow-OT | TabbyFlow-VP |
|---|---|---|---|---|
| Utility | 0.5358±0.0058 | 0.4801±0.0049 | 0.5333±0.0206 | 0.5622±0.0213 |
| Risk | 0.0951±0.0097 | 0.0791±0.0090 | 0.1581±0.0058 | 0.1392±0.0051 |
| WD | 2.5976±0.0025 | 2.7618±0.0026 | 2.3773±0.0028 | 2.4788±0.0124 |
| Shape | 3.33±0.03 | 5.92±0.03 | 1.93±0.03 | 1.45±0.04 |
| Trend | 5.10±0.10 | 8.85±0.09 | 6.13±0.18 | 5.54±0.18 |
| $\alpha$-precision | 93.77±0.13 | 88.2±0.14 | 93.85±0.15 | 97.72±0.14 |
| $\beta$-recall | 37.01±0.17 | 27.82±0.14 | 50.06±0.17 | 45.81±0.18 |

### G.2.2 High Dimensional Data: Gene Expression

Table 24 reports a high-dimensional stress test on Gene Expression of Cancer dataset ($n = 801$, $D_{\text{num}} = 20,264$, $D_{\text{cat}} = 1$) [4]. To avoid OOM in the Transformer-VAE, we first project numerical features to 256 dimensions with layer normalisation, enabling TabSynFlow to learn flow matching in a compact latent space, while TabbyFlow learns a data-space variational flow with an MLP. Since $D_{\text{cat}} = 1$, utility uses univariate ROC only and we report Utility $= (\text{ROC}_{\text{uni}} + \text{CIO})/2$; we omit Trend due to its $O(D^2)$ cost, but CIO still captures multivariate dependence through regression relationships. In this regime, TabSynFlow achieves substantially higher utility, with near-zero disclosure risk across settings. Meanwhile TabbyFlow could preserve global fidelity, indicated with lower WD than TabSynFlow. In this stress test, latent-space FM scales more favorably than data-space FM with an MLP backbone in providing practical synthetic data. The comparison highlights complementary strengths: latent-space FM benefits from dimensionality reduction that stabilises transport, while data-space FM avoids representation learning but requires sufficient model capacity to represent high-dimensional velocity fields. In the $> 20K$ setting, an MLP backbone appears insufficient, suggesting architecture is the bottleneck rather than the TabbyFlow formulation itself.

Table 24: High dimensional stress test on the gene expression dataset ($D_{\text{num}} = 20264$, $D_{\text{cat}} = 1$): comparison of latent-space flow matching (TabSynFlow) and data-space flow matching (TabbyFlow) under OT and VP trajectories. We report the mean $\pm$ standard deviation across 20 seeds of synthetic data generation. The results show that TabSynFlow substantially outperforms TabbyFlow on Utility, and Shape in this ultra-high-dimensional regime, while TabbyFlow achieves lower WD over TabSynFlow. Overall, latent-space FM provides a more favorable utility trade-off at >20K dimensions, even with a 256-d projection to avoid OOM, while TabbyFlow preserves joint distribution.

| Evaluation | TabSynFlow-OT | TabSynFlow-VP | TabbyFlow-OT | TabbyFlow-VP |
|---|---|---|---|---|
| Utility | 0.8233±0.0133 | 0.8289±0.0133 | 0.6691±0.0153 | 0.6714±0.0103 |
| Risk | 0.0±0.0 | 0.0033±0.0148 | 0.0±0.0 | 0.0002±0.0008 |
| WD | 212.60±1.04 | 206.90±1.17 | 149.80±0.71 | 155.36±0.62 |
| Shape | 9.90±0.08 | 9.22±0.10 | 20.21±0.21 | 20.47±0.18 |

### G.3 Architecture Comparison of TabbyFlow: MLP vs. Transformers

Table 25 compares TabbyFlow with MLP vs. Transformer backbones under OT and VP trajectories (mean±std across seeds). Overall, the MLP backbone is competitive with Transformers and often stronger on Utility. In particular, MLP–OT is best on UK, Indonesia, and Canada, while Transformers provide a

---

[4]We removed columns that has zeros on all observations. Data can be accessed in `https://archive.ics.uci.edu/dataset/401/gene+expression+cancer+rna+seq`

Table 25: Performance of TabbyFlow when using MLP and transformers architecture. Note that in here we did not use gradual decrease Gaussian loss weight like implemented by Guzmán-Cordero et al. (2025), following our default implementation. We report the mean ± standard deviation across 20 generation seeds. The results highlights the competitiveness of MLP compared to Transformers in producing synthetic data.

| Evaluation | Data | MLP-OT | MLP-VP | Transformers-OT | Transformers-VP |
|---|---|---|---|---|---|
| Utility | UK | 0.8333±0.0191 | 0.8066±0.0197 | 0.7809±0.0172 | 0.7917±0.0139 |
| | Canada | 0.7892±0.0178 | 0.7472±0.0188 | 0.7854±0.0129 | 0.7397±0.0187 |
| | Fiji | 0.7263±0.0205 | 0.7451±0.0216 | 0.7391±0.0172 | 0.7394±0.0193 |
| | Rwanda | 0.7284±0.0231 | 0.7358±0.0185 | 0.7442±0.0270 | 0.7209±0.0225 |
| | Indonesia | 0.9191±0.0139 | 0.8473±0.0372 | 0.9124±0.0089 | 0.8837±0.0281 |
| | Adult | 0.7720±0.0231 | 0.7581±0.0165 | 0.5637±0.0113 | 0.7437±0.0156 |
| | Churn | 0.7966±0.0098 | 0.8066±0.0102 | 0.8738±0.0161 | 0.8339±0.0095 |
| Risk | UK | 0.4889±0.0043 | 0.4765±0.0072 | 0.4716±0.0054 | 0.4820±0.0040 |
| | Canada | 0.3373±0.0156 | 0.3008±0.0148 | 0.3350±0.0164 | 0.3209±0.0185 |
| | Fiji | 0.5705±0.0063 | 0.5588±0.0074 | 0.5727±0.0091 | 0.5606±0.0048 |
| | Rwanda | 0.5379±0.0139 | 0.5180±0.0105 | 0.5359±0.0098 | 0.5306±0.0130 |
| | Indonesia | 0.6556±0.0133 | 0.6447±0.0096 | 0.6565±0.0146 | 0.6546±0.0138 |
| | Adult | 0.5443±0.0177 | 0.5037±0.0172 | 0.5432±0.0139 | 0.5609±0.0153 |
| | Churn | 0.0773±0.0579 | 0.0753±0.0708 | 0.1249±0.0904 | 0.1427±0.0736 |

clear gain mainly on Churn (and a small gain on Rwanda). On Adult, Transformer-OT underperforms markedly, suggesting higher sensitivity to training details. Disclosure Risk differences are generally small across backbones, with Churn showing higher variance. Given comparable performance and substantially lower compute cost, we use the MLP backbone as the default and report Transformer results as an ablation.

