# OpenReview forum: "Flow Matching for Tabular Data Synthesis"
_TMLR — Accepted by TMLR_

### Review · Reviewer_mE5c · 2026-01-03

**Summary Of Contributions:**

The manuscript presents a benchmark of flow matching methods for tabular data, examining the impact of various design choices on these methods, including architecture design (flow matching in latent space or data space), specification of the vector field, number of function evaluations, and the distinction between deterministic and stochastic integration for sampling. The benchmark consists of:
3 methods, TabDDPM, TabSyn, TabsynFlow, and TabbyFlow, and different flavours of trajectory specification (Variance Preserving v. OT based)
7 datasets, of which 5 are census-type datasets (with a mix of continuous or categorical variables), and 2 are from the standard UCI benchmark.
2 metrics, termed "Utility" and "Risk", which correspond to an aggregate score of additional metrics.

**Audience:**

No

**Audience Explanation:**

The empirical evaluation provides a solid foundation, though expanding the benchmark with additional datasets and evaluation metrics would strengthen its contribution to the broader machine learning community. For instance, the recent work on continuous diffusion for mixed-type tabular data (Mueller et al., ICLR 2025) introduces a comprehensive set of metrics that effectively capture the quality of generative modeling. Incorporating similar metrics here would provide a more complete picture of the proposed methods' capabilities.

**Claims And Evidence:**

No

**Claims Explanation:**

The benchmark offers interesting insights into the behavior of various formulations of flow-based models, but it is quite specific, as it focuses on a very specific type of tabular data: census datasets. In particular:

- It uses metrics that are not commonly used in machine learning, but seem to be bespoke to the specific field of applications. For a benchmark of generative models for tabular data, I would expect to see some standard metrics, such as Wasserstein or MMD, as well as a detailed analysis of the models' ability to capture the summary statistics of the various variables. Additionally, at least a sketch of the metrics and a brief description should be present in the main text, not just in the appendix, especially in this case, where the metrics are highly specialized to the field of application.

- The datasets used are diverse in terms of the number of variables and size, but are too specific to the field of applications and also insufficient for providing an exhaustive picture of the performance of generative models on tabular data. For example, how does the model perform with categorical variables with high cardinality (100+ categories like zip codes)?

- Additional architecture ablations would be useful, e.g., how does data-space vs. latent-space flow matching compare on tables of hundreds or thousands of variables (for instance, a genomics dataset)? Additionally, would a transformer-based architecture for TabbyFlow improve the performance?

**Requested Changes:**

1. Evaluation metrics: Include standard generative modeling metrics (e.g., Wasserstein distance, MMD) alongside the domain-specific utility/risk measures. The metrics reported in Mueller et al. (ICLR 2025) for continuous diffusion on mixed-type tabular data provide a useful reference. Additionally, please include at least a brief description of the evaluation metrics in the main text, rather than relegating them entirely to the appendix.

2. Dataset diversity: Expand the benchmark beyond census data to include datasets from other domains. In particular, it would be valuable to evaluate performance on:
  1. Categorical variables with high cardinality (100+ categories, e.g., zip codes)
  2. Tables with substantially more variables (hundreds or thousands of features). This, in particular, would provide a useful evaluation of the tradeoff between data-space and latent-space modeling for flow-based generative modeling, a duality that has been extensively explored in other domains, such as image generation.

---

> ### Author Response · Authors · 2026-01-17
> **We address metric standardization, dataset diversity, and architecture ablations. Added standard metrics (Wasserstein, MMD, C2ST, α-precision, β-recall), high-cardinality stress tests (~700+ categories), and high-dimensional tests (D approx 20K). Clarified latent-space FM scalability and added Transformer vs. MLP ablations for TabbyFlow.**
>
> We thank the reviewer for the careful reading and constructive suggestions. We address each point below and indicate where the revision is reflected in the manuscript. The revisions are highlighted in blue.
>
> ### Summary of Contributions
>
> **Response:**
>
> Thank you for the summary. We confirm the benchmark scope as stated: we compare TabDDPM, TabSyn, TabSynFlow, and TabbyFlow under multiple design choices (representation, trajectory, and dynamics), on seven datasets (five census-type, two standard tabular benchmarks), and evaluate using utility and disclosure risk as primary criteria.
>
> ### Metrics are bespoke, please include standard metrics and define them in the main text
>
> **Response:**
>
> We addressed this review by (i) providing concise definitions of Utility and Risk into the main text, including the aggregation rule and the evaluation protocol [Section 4.3.1 and 4.3.2, details in Appendix D.1 and D.2], and (ii) adding standard distributional fidelity and detectability diagnostics, including marginal Wasserstein distance and maximum mean discrepancy, and low-order shape and trend statistics, a classifier two-sample test (C2ST) detection score, and sample-level quality evaluation [Appendix D.3 with detailed results in Appendix E.1]. We report Utility and Risk in the main text as the primary benchmark evaluations and provide the full set of diagnostics and definitions in the appendix to avoid over-interpreting any single metric.
>
> ### Dataset diversity, high-cardinality categorical variables (100+ categories) and high-dimensional data
>
> **Response:**
>
> This is an interesting Ablation we found so far. Thank you for suggesting this. We addressed the concern by (i) reporting dataset statistics including maximum single-column cardinality [Section 4.1, Table 1] (ii) adding ablation on high cardinality data (Diabetes, url: https://archive.ics.uci.edu/dataset/296/diabetes+130-us+hospitals+for+years+1999-2008) [Appendix G.2.1, Table 23] and high dimensional data (Cancer gene expression, url: https://archive.ics.uci.edu/dataset/401/gene+expression+cancer+rna+seq) [Appendix G.2.2, Table 24].
>
> ### Architecture ablations: Transformer backbone for TabbyFlow
>
> **Response:**
>
> Our benchmark controls for model capacity by defaulting to an MLP backbone for FM and diffusion components to ensure comparability with MLP-based baselines, while latent-space methods retain their Transformer-based VAE components. To address this concern, we add an explicit TabbyFlow ablation comparing MLP versus Transformer backbones under both trajectory settings [Appendix G.3, Table 25].
>
> ### TMLR audience interest
>
> **Response:**
>
> Our study targets a setting where generative tabular modeling is evaluated under a utility-disclosure-risk relation, which is central in official statistics and relevant to other sensitive tabular domains and complementary to existing ML literature on tabular data generation. To address the concern of increasing interest and connection to the ML literature, we add standard diagnostic metrics in tabular generative modelling [Appendix E.1]. We also add ablation in other privacy-sensitive tabular settings beyond our core benchmark in [Appendix G.2].

---

### Review · Reviewer_WQPK · 2026-01-09

**Summary Of Contributions:**

The paper presents an experimental comparison of generative approaches for tabular data, starting from diffusion-based benchmark methods. The thesis is that flow matching-based approaches, in particular conditional flow matching and variational flow matching, can outperform diffusion-based ones. A core concept the authors try to convey is that among the possible variations of genAI for tabular data, a central role is the particular choice one makes for four axes: representation (latent space or not), learning target (CFM or Variational), trajectory (optimal transport or variance preserving), and dynamics (ODE or SDE). Hence, they focus on two possible combinations of these: TabSynFlow (ODE, latent space, and CFM), introduced by them, and TabbyFlow (SDE, data space, and Variational), which already exists in the literature. Hence, one possible weakness of the paper is that from the theoretical point of view, they make a choice to introduce a single new method, but limited theoretical explanation on why one should expect one combination of the four axes to be better than another is provided.
That said, the paper tries to collocate in an experimental fashion for a detailed comparison between methods, which sounds readable and complete. As minor issue, the origin of the error bars is not provided either apparently in the text, at least not in the table where they appear.

In conclusion, given the accent and novelty is mainly regarding the experimental part, it would possibly be better to cut down on the theoretical part and keep just some highlights of flow matching, being well known in the community nowadays.

**Audience:**

Yes

**Audience Explanation:**

The generation of tabular data with top performance genAI methods like diffusion models and flow matching is a current challenge in the community since the data space is discrete, and it is not that immediate to extend standard generative approaches to such domain.

**Broader Impact Concerns:**

I require the addition of a Broader Impact Statement since privacy is a fundamental characteristic of census data, as already partially discussed in the text regarding the RISK metric.

**Claims And Evidence:**

Yes

**Claims Explanation:**

I would say partially yes, since the methods and the experiments are well explained, but key points like the error bars and the computational resources that have been used are not completely explained in this version.

**Requested Changes:**

- explain the error bars
- possibly cut on the theory and expand the experimental section, moving some results from the appendix to main text

Minor:
- the sentence "Our most interesting finding is while it is not commonly used in FM studies, using
VP path in TabbyFlow has potential to produce better optimised synthetic data with higher utility but lower
disclosure risk." is a bit intricate, and could be rephrased

---

> ### Author Response · Authors · 2026-01-17
> **We thank the reviewer for the thoughtful feedback, especially on reproducibility and presentation. We address each request below and indicate where revisions were made. The revisions are highlighted in blue.**
>
> We thank the reviewer for the thoughtful feedback, especially on reproducibility and presentation. We address each request below and indicate where revisions were made. The revisions are highlighted in blue.
>
> ### Summary of Contributions and overall framing
>
> **Response:**
>
> Thank you for the clear summary. We agree that the novelty of the work is primarily empirical. In the revision, we streamline the theory background to highlight only what is necessary to understand the benchmark axes and experimental design, and we move non-essential theoretical exposition to the appendix such as trajectory setting [Appendix F.2] and TabSyn [Appendix B] so that the main text prioritises the flow matching in latent and data space.
>
> ### Explain the error bars
>
> **Response:**
>
> We addressed this concern by clarifying the caption of each relevant table and figure that error bars represent variability across random seeds. Concretely, results are reported as mean $\pm$ standard deviation across 20 (seeds) independent data synthesis in all tables.
>
> ### Make the code available and ensure full reproducibility
>
> **Response:**
>
> We addressed this concern by attaching the code for reviewers in the supplementary materials on this submission and we will make it public upon upon acceptance. We also add a reproducibility statement in the paper specifying software versions, hardware we used in Appendix D.4.
>
> ### Cut theory and expand experiments, move key results from appendix to main text
>
> **Response:**
>
> We addressed the reviewer’s concern about the content by moving some key definitions and add the experiment details into the main text so that the benchmark can be interpreted without relying on the appendix. We also move non-essential theoretical exposition to the appendix such as trajectory setting [Appendix F.2], TabSyn [Appendix A.2] so that the main text prioritises the flow matching in latent and data space and make the focus the main paper on the benchmark protocol, metrics, and findings. We also moved the additional data results (ID, AD, and CH) into [Section 5.1].
>
> ### Minor comment on phrasing (intricate sentence)
>
> **Response:**
>
> Thank you. We rewrite the sentence to make the claim direct and easier to parse, without changing its meaning.
>
> ### Broader Impact Statement
>
> **Response:**
>
> We adressed this by adding a concise broader impact statement that highlights both potential benefits (data access under governance) and risks (privacy leakage and invalid conclusions), and stresses dataset-specific risk assessment.

---

### Review · Reviewer_hYDC · 2026-01-09

**Summary Of Contributions:**

The paper proposes a benchmark of synthetic data generation for tabular data in the context of census data, based on flow matching (FM) and diffusion.
The main challenges in generative models are data quality, diversity, and privacy preservation; when applied to tabular data, handling a mix of categorical and numerical feature is an additional challenge.

Many factors impact the performance of FM models; the paper's benchmark tries to disentangle some of them.
In particular, it provides comparisons between:
- FM and diffusion models (namely TabDDPM and TabSyn).
- FM models in the variable or latent space (TabbyFlow vs TabSynFlow)
- optimal transport (OT) and Variance Preserving (VP) probability *couplings* in FM (see Required clarifications)
- deterministic and stochastic FM samplers

**Audience:**

Yes

**Audience Explanation:**

I tend to answer positively, but I am not 100% sure: the paper's motivation is about census data, and thus limited to 4 dataset in this setting. How common in the tabular data literature (beyond census data, e.g. genomics) is the concern about data disclosure?

**Broader Impact Concerns:**

None.

**Claims And Evidence:**

No

**Claims Explanation:**

- Given the extremely high number of hyperparameters to choose, I am not 100% confident about the validity of the conclusions that are drawn. This is not a flaw of the paper per se, the design space combinatorial nature is simply too big too explore properly. However, is is possible to really improve reproducibility by releasing an open source benchmark repository, where various configurations and methods could be tested by other researchers even after the release of this paper. This is for example done in this TMLR paper: https://arxiv.org/abs/2407.11676
- The architecture choice for the velocities in FM is a vanilla MLP, as in Zhang 2024. It is known in the FM literature that this choice is critical. Since the paper's contribution is a benchmark, the authors could provide a more thorough discussion about this choice of architecture.
- the metrics are "data utility" and "risk disclosure": from the cited papers, there are multiple methods to measure these quantities, and i could not find a precise definition in the paper. This evaluation of generative models is a notoriously hard question (see e.g. in the imaging community, discussions about the Fréchet Inception Distance: https://arxiv.org/abs/2306.04675).
Since the paper's contribution is a benchmark, the metrics should be introduced in depth in the main part of the paper.

**Requested Changes:**

**Please use a different color to highlight changes in revision**
- There is a major confusion in the paper: the authors mention that linear interpolation is OT trajectory (sec 2.3) ; this is not true. OT (Tong et al 2024) refers to using minibatch OT coupling when sampling pairs of variables $(x_0, x_1)$. Empirically, this provides straighter **unconditional** path (thus making the ODE resolution easier). But this is not related to taking linear interpolant for the **conditional** paths.
This whole paragraph needs to be rewritten, e.g. sentences such as "[VP]  In this approach, noise is gradually added to the data during the forward process, while the reverse process learns to de-noise using time-dependent vector fields" are in fact true for any flow matching approach.
"Unlike OT paths, VP diffusion trajectories evolve smoothly in both magnitude and direction over time.": are OT paths not smooth in time and space? They are linear in both.
- the presentation can be simplified, ie in 2.15 the authors introduce $\sigma(t)$, but it is in fact equal to $t$ in TabSyn, so this expression should be plugged equation 2.15 to make it simpler.
- the authors claim that categorical features are a challenge, but doesn't using a continuous latent space circumvent this difficulty?
- "offering a continuum between precise transport and exploratory regularisation": why is an ODE a "precise trnasport" and an SDE "exploratory regularization"? In which sense does an SDE provide regularization? Can the authors provide references?
- Variational FM being way less standard than FM, the Mean Field approximation should be introduced more thoroughly (how is mean field related to conditional independence across dimensions?) at the bottom of page 3. Why is $q^\theta$ called a *mean-field* posterior approximation in page 6?
- The one hot encoding approach of TabbyFlow seems very costly, are there some alternative? In particular, there now exist discrete flow matching approaches ("Discrete Flow Matching", Gat et al 2024). Is it possible to mix continuous and discrete flow matching to generate a mix of continuous and categorical data ?
- In fig 1 I can't understand while there are both dotted and solid lines on each graphs, while the left one has title "utility" and thus should have only solid lines?

## Cosmetic
- Use appropriate citation format when the citation is not aprt of the sentence, eg  " the same marginal distribution pt(x) Eijkelboom et al. (2024):" should be " the same marginal distribution pt(x) (Eijkelboom et al., 2024)." See the use of \citet vs \citep in the natbib package.
- why use "pseudo-time" to refer to 't' instead of simply "time", as common in the literature?
- the quality of Figure 1 coudl be improved: the labels of the dashed lines are hardly readable, the figure on the right could have a smaller range of y values to better highlight

---

> ### Author Response · Authors · 2026-01-17
> **We commit to open-source benchmark repository with reproducibility scripts upon acceptance. Add MLP vs. Transformer ablations and clarify metric definitions (Utility and Risk) with equations in main text. Clarify OT terminology, mean-field assumptions, and pragmatic role of latent-space modeling.**
>
> We thank the reviewer for the feedback. We address the major concerns on terminology, reproducibility, evaluation, and modeling choices. Responses include a placeholder to show where the manuscript was revised. The revisions are highlighted in blue.
>
> ### Large hyperparameter space and reproducibility
>
> **Response:**
>
> We addressed this concern by providing a code file for reviewers in the supplementary materials on this submission. Once this paper is accepted, we will release the code openly on GitHub.
>
> ### Architecture choice (vanilla MLP) needs deeper discussion
>
> **Response:**
>
> In the benchmark, we default to an MLP backbone to control model capacity across FM and diffusion baselines, as revised in Section 4.2. Latent-space methods still use Transformer-based VAE components for encoding and decoding. To directly address the concern of architectural sensitivity, we add an explicit ablation for TabbyFlow comparing MLP and Transformer backbones in Appendix G.3.
>
> ### Metric definitions (Utility and disclosure Risk) should be introduced in the main text
>
> **Response:**
>
> We addressed this concern by adding more explanation of Utility and Risk into the main text [Section 4.3], while keeping implementation details and pseudocode in the appendix [Appendix D]. We also add standard diagnostic metrics (distributional and detectability) to complement the domain-oriented utility and risk criteria.
>
> ### Clarification: OT trajectory versus linear interpolation
>
> **Response:**
>
> We moved this section into [Appendix F.2]
> We agree that our wording could be interpreted ambiguously given that “OT” is used in different ways across the literature. In our manuscript, we use “OT path” in the sense introduced in the Flow Matching framework of Lipman et al. (2022), namely the straight (affine) conditional probability path (linear interpolation in the conditional mean, with a simple variance schedule). To avoid confusion with minibatch OT coupling for endpoint pairing (as discussed in other works), we remove the citation to Tong et al. (2024).
>
> We also revise the VP-path description. In particular, we remove the incorrect smoothness contrast and instead clarify that in the VP construction the conditional noise level is controlled by the schedule that makes trajectories can appear to change slowly at the beginning and more rapidly near the end (t $\approx$ 1), which is different than OT.
>
> ### the presentation in TabSyn can be simplified
>
> We addressed this concern by revising the expalantion in TabSyn [now in Appendix B] into:
>
> Given linear noise schedule t, the corresponding reverse process is an SDE:
>
> $$
> \mathrm{d} z_t = -2 t \nabla_z \log p_t(z_t) \, \mathrm{d}t + \sqrt{2t} \, \mathrm{d}\omega_t.
> $$
>
> ### Why is an ODE a "precise transport" and an SDE "exploratory regularization"?
>
> We only meant that ODE sampling is deterministic (a fixed mapping given an initial noise), while SDE sampling injects stochasticity along the trajectory. We addressed this concern by revising the text to describe this as deterministic versus stochastic sampling.
>
> ### Categorical features are a challenge and latent space circumvents it?
>
> While latent-space modeling reduces the difficulty of applying continuous-time FM to mixed-type
> tables, it does not eliminate the categorical challenge. Further, categorical structure should still be represented and decoded from latent variables, and maintaining calibrated category probabilities and rare-category coverage remains non-trivial, especially under high-cardinality and strong cross-column dependencies.
>
> ### Mean-field approximation in VFM needs clearer explanation
>
> **Response:**
>
> The VFM paper by Eijkelboom et al. (2024) proves formally that mean-field solution is sufficient to learn VFM. We also addressed the reviewer's concern by explicitly defining in Section 2.2 that the mean-field variational family as a factorised posterior approximation $q_t^\theta(x_1\mid x_t)=\prod_{d=1}^D q_t^\theta(x_1^d\mid x_t)$,
> which corresponds to conditional independence across dimensions under the variational posterior (not under the true posterior).
>
> ### One-hot encoding cost and alternatives, relation to discrete flow matching
>
> **Response:**
>
> In the benchmark, we follow the common practices of tabular generative models, including baseline methods, which is using one hot encoding. On the other hand, Discrete Flow Matching is a promising direction for discrete state spaces, and hybrid continuous--discrete modeling for mixed-type tables is an important future direction, but it introduces additional design choices beyond the scope of this benchmark. We also addressed this concern by clarifying this in Section 6.2.
>
> ### Figure 1 line styles, readability, and Cosmetic and writing corrections
>
> **Response:**
>
> We addressed this concern by revising the caption and legend to clearly indicate which line style corresponds to which trajectory, and we improve figure readability by enlarging labels and adjusting the axis range where appropriate.

---

### Author Response · Authors · 2026-02-03
**Consolidated revision uploaded (please refer to latest PDF)**

Dear Action Editor and reviewers,

We thank you for the detailed reviews and discussion.

We uploaded an updated revision on February 1st and a follow-up consolidated revision on February 3rd. Please use the latest PDF as the reference version.

Summary of changes:
- Updated the computation of Wasserstein distance and MMD to the joint multivariate setting instead of separate between continuous and categorical. Due to memory constraints, we compute these metrics in batches (n=5,000) and report the mean across batches.
- The upload on February 3rd contains typo and wording fixes only, with no impact on experiments or conclusions.

Other changes follow the reviewers’ comments and are reflected in the revised manuscript and our response. We will keep the manuscript stable from this point unless requested by the Action Editor.

Thank you for your time. We are happy to clarify any remaining points.

---

### Decision · Action_Editor_5shB · 2026-03-04

**Recommendation:** Accept as is

**Audience:**

Yes

**Audience Explanation:**

Using generative modelling techniques for synthetic tabular data generation is certainly a topic that is of some interest in the TMLR readership. As mentioned above, the focus of the paper is in fact somewhat narrower than this, by targeting a mix of distributional fidelity and statistical non-disclosure guarantees but it doesn't read like a niche paper either.

**Claims And Evidence:**

Yes

**Claims Explanation:**

The paper's main contribution is an empirical benchmark of synthetic data generation for tabular data using flow matching and diffusion techniques.

- The methods tested in the benchmark are timely and adequately described in the text (with some content moved to appendix in the revised version, as a result of interactions with reviewers).
- The data is somewhat specific (census data) but probably representative of small-scale tabular data (appendix G reports other results on different types of data) and most importantly the metrics used for the benchmark are sound and well-described in the text. A distinctive feature here is that these metrics also include disclosure (or privacy) statistics that are specific to categorical variables. While this concern is most relevant for census data, it can be meaningful as well in other settings where the tables correspond to personal data.

The caveat in such benchmarks, as mentioned by one of the reviewers, is that there are so many design parameters that it is hard to be sure of the validity of the conclusions drawn from the experiment. This being said, the authors' experiment are carefully designed -focusing on a selection of design options (Sections 3.1) that are relevant- and the results are precisely reported, which overall result in a valid experimental contribution for TMLR.